# Genomics of 1 million parent lifespans implicates novel pathways and common diseases and distinguishes survival chances

Paul RHJ Timmers[1], Ninon Mounier[2,3], Kristi Lall[4,5], Krista Fischer[4,5], Zheng Ning[6], Xiao Feng[7], Andrew D Bretherick[8], David W Clark[1], eQTLGen Consortium, Xia Shen[1,6,7], Tõnu Esko[4,9], Zoltán Kutalik[2,3], James F Wilson[1,8], Peter K Joshi[1,2]*

[1]Centre for Global Health Research, Usher Institute of Population Health Sciences and Informatics, University of Edinburgh, Edinburgh, United Kingdom; [2]Institute of Social and Preventive Medicine, University Hospital of Lausanne, Lausanne, Switzerland; [3]Swiss Institute of Bioinformatics, Lausanne, Switzerland; [4]Estonian Genome Center, Institute of Genomics, University of Tartu, Tartu, Estonia; [5]Institute of Mathematics and Statistics, University of Tartu, Tartu, Estonia; [6]Department of Medical Epidemiology and Biostatistics, Karolinska Institutet, Stockholm, Sweden; [7]State Key Laboratory of Biocontrol, School of Life Sciences, Sun Yat-sen University, Guangzhou, China; [8]MRC Human Genetics Unit, Institute of Genetics and Molecular Medicine, University of Edinburgh, Edinburgh, United Kingdom; [9]Broad Institute of Harvard and MIT, Cambridge, United States

**Abstract** We use a genome-wide association of 1 million parental lifespans of genotyped subjects and data on mortality risk factors to validate previously unreplicated findings near *CDKN2B-AS1*, *ATXN2/BRAP*, *FURIN/FES*, *ZW10*, *PSORS1C3*, and 13q21.31, and identify and replicate novel findings near *ABO*, *ZC3HC1*, and *IGF2R*. We also validate previous findings near 5q33.3/*EBF1* and *FOXO3*, whilst finding contradictory evidence at other loci. Gene set and cell-specific analyses show that expression in foetal brain cells and adult dorsolateral prefrontal cortex is enriched for lifespan variation, as are gene pathways involving lipid proteins and homeostasis, vesicle-mediated transport, and synaptic function. Individual genetic variants that increase dementia, cardiovascular disease, and lung cancer – but not other cancers – explain the most variance. Resulting polygenic scores show a mean lifespan difference of around five years of life across the deciles.

**Editorial note:** This article has been through an editorial process in which the authors decide how to respond to the issues raised during peer review. The Reviewing Editor's assessment is that all the issues have been addressed (see decision letter).

DOI: https://doi.org/10.7554/eLife.39856.001

*For correspondence:
peter.joshi@ed.ac.uk

Group author details:
eQTLGen Consortium See page 31

## Introduction

Human lifespan is a highly complex trait, the product of myriad factors involving health, lifestyle, genetics, environment, and chance. The extent of the role of genetic variation in human lifespan has been widely debated (*van den Berg et al., 2017*), with estimates of broad sense heritability ranging from around 25% based on twin studies (*Ljungquist et al., 1998*; *Herskind et al., 1996*; *McGue et al., 1993*) (perhaps over-estimated [*Young et al., 2018*]) to around 16.1%, (narrow sense

**eLife digest** Ageing happens to us all, and as the cabaret singer Maurice Chevalier pointed out, "old age is not that bad when you consider the alternative". Yet, the growing ageing population of most developed countries presents challenges to healthcare systems and government finances. For many older people, long periods of ill health are part of the end of life, and so a better understanding of ageing could offer the opportunity to prolong healthy living into old age.

Ageing is complex and takes a long time to study – a lifetime in fact. This makes it difficult to discern its causes, among the countless possibilities based on an individual's genes, behaviour or environment. While thousands of regions in an individual's genetic makeup are known to influence their risk of different diseases, those that affect how long they will live have proved harder to disentangle. Timmers et al. sought to pinpoint such regions, and then use this information to predict, based on their DNA, whether someone had a better or worse chance of living longer than average.

The DNA of over 500,000 people was read to reveal the specific 'genetic fingerprints' of each participant. Then, after asking each of the participants how long both of their parents had lived, Timmers et al. pinpointed 12 DNA regions that affect lifespan. Five of these regions were new and had not been linked to lifespan before. Across the twelve as a whole several were known to be involved in Alzheimer's disease, smoking-related cancer or heart disease. Looking at the entire genome, Timmers et al. could then predict a lifespan score for each individual, and when they sorted participants into ten groups based on these scores they found that top group lived five years longer than the bottom, on average.

Many factors beside genetics influence how long a person will live and our lifespan cannot be read from our DNA alone. Nevertheless, Timmers et al. had hoped to narrow down their search and discover specific genes that directly influence how quickly people age, beyond diseases. If such genes exist, their effects were too small to be detected in this study. The next step will be to expand the study to include more participants, which will hopefully pinpoint further genomic regions and help disentangle the biology of ageing and disease.

DOI: https://doi.org/10.7554/eLife.39856.002

12.2%) based on large-scale population data (*Kaplanis et al., 2018*). One very recent study suggests it is much lower still (<7%) (*Ruby et al., 2018*), pointing to assortative mating as the source of resemblance amongst kin.

Despite this modest heritability, extensive research has gone into genome-wide association studies (GWAS) finding genetic variants influencing human survival, using a variety of trait definitions and study designs (*Deelen et al., 2011*; *Sebastiani et al., 2012*; *Beekman et al., 2013*; *Broer et al., 2015*; *Joshi et al., 2016*; *Pilling et al., 2016*; *Zeng et al., 2016*; *Pilling et al., 2017*). GWAS have primarily focused on extreme cases of long-livedness (longevity) – individuals surviving past a certain age threshold – and scanning for differences in genetic variation from controls. While this case-control design has the advantage of focusing on highly statistically-informative individuals, who also often exhibit extreme healthspan and have potentially unique genetic attributes (*Sebastiani et al., 2013*; *Sebastiani et al., 2016*), the exceptional nature of the phenotype precludes collection of large samples, and differences in definitions of longevity complicate meta-analysis. As a result, only two robustly replicated, genome-wide significant associations (near *APOE* and *FOXO3*) have been made to date (*Broer et al., 2015*; *Deelen et al., 2014*).

An alternative approach is to study lifespan as a quantitative trait in the general population and use survival models (such as Cox proportional hazards [*Cox, 1972*]) to allow long-lived survivors to inform analysis. However, given the incidence of mortality in middle-aged subjects is low, studies have shifted to the use of parental lifespans with subject genotypes (an instance of Wacholder's kin-cohort method [*Wacholder et al., 1998*]), circumventing the long wait associated with studying age at death in a prospective study (*Joshi et al., 2016*; *Pilling et al., 2016*). In addition, the recent increase in genotyped population cohorts around the world, and in particular the creation of UK Biobank (*Bycroft et al., 2017*), has raised GWAS sample sizes to hundreds of thousands of individuals, providing the statistical power necessary to detect genetic effects on mortality.

A third approach is to gather previously published GWAS on risk factors thought to possibly affect lifespan, such as smoking behaviour and cardiovascular disease (CVD), and estimate their actual independent, causal effects on mortality using Mendelian Randomisation. These causal estimates can then be used in a Bayesian framework to inform previously observed SNP associations with lifespan (*McDaid et al., 2017*).

Here, we blend these three approaches to studying lifespan and perform the largest GWAS on human lifespan to date. First, we leverage data from UK Biobank and 26 independent European-heritage population cohorts (*Joshi et al., 2017*) to carry out a GWAS of parental survival, quantified using Cox models. We then supplement this with data from 58 GWAS on mortality risk factors to conduct a Bayesian prior-informed GWAS (iGWAS). Finally, we use publicly available case-control longevity GWAS statistics to compare the genetics of lifespan and longevity and provide collective replication of our lifespan GWAS results.

We also examine the diseases associated with lifespan-altering variants and the effect of known disease variants on lifespan, to provide insight into the interplay between lifespan and disease. Finally, we use our GWAS results to implicate specific genes, biological pathways, and cell types, and use our findings to create and test whole-genome polygenic scores for survival.

## Results

### Genome-wide association analysis

We carried out GWAS of survival in a sample of 1,012,240 parents (60% deceased) of European ancestry from UK Biobank and a previously published meta-analysis of 26 additional population cohorts (LifeGen [*Joshi et al., 2017*]; *Table 1—source data 1*). We performed a sex-stratified analysis and then combined the allelic effects in fathers and mothers into a single parental survival association in two ways. First, we assumed genetic variants with common effect sizes (CES) for both parents, maximising power if the effect is indeed the same. Second, we allowed for sex-specific effect sizes (SSE), maximising power to detect sexually dimorphic variants, including those only affecting one sex. The latter encompasses a conventional sex-stratified analysis, but uses only one statistical test for the much more general alternative hypothesis that there is an effect in at least one sex.

We find 12 genomic regions with SNPs passing genome-wide significance for one or both analyses (p < 2.5 × 10⁻⁸, accounting for the two tests CES/SSE) (*Figure 1*; *Table 1*). Among these are five loci discovered here for the first time, at or near *MAGI3*, *KCNK3*, *HTT*, *HP*, and *LDLR*. Carrying one copy of a life-extending allele is associated with an increase in lifespan between 0.23 and 1.07 years (around 3 to 13 months). Despite our sample size exceeding 1 million phenotypes, a variant had to have a minor allele frequency exceeding 5% and an effect size of 0.35 years of life or more per allele for our study to detect it with 80% power.

We also attempted to validate novel lifespan SNPs discovered by *Pilling et al. (2017)* in UK Biobank at an individual level by using the LifeGen meta-analysis as independent replication sample. Testing 20 candidate SNPs for which we had data available, we find directionally consistent, nominally significant associations for six loci (p < 0.05, one-sided test), of which three have sex-specific effects. We also provide evidence against three putative loci but lack statistical power to assess the remaining 11 (*Figure 2*, *Figure 2—source data 1*).

We then used our full sample to test six candidate SNPs previously associated with longevity (*Zeng et al., 2016*; *Deelen et al., 2014*; *Flachsbart et al., 2009*; *Sebastiani et al., 2017*) for association with lifespan, and find directionally consistent evidence for SNPs near *FOXO3* and *EBF1*. The remaining SNPs did not associate with lifespan despite apparently adequate power to detect any effect similar to that originally reported (*Figure 2*, *Figure 2—source data 1*).

Finally, we tested a deletion, d3-GHR, reported to affect male lifespan by 10 years when homozygous (*Ben-Avraham et al., 2017*) by converting its effect size to one we expect to observe when fitting an additive model. We used a SNP tagging the deletion and estimated the expected effect size in a linear regression for the (postulated) recessive effect across the three genotypes, given their frequency (see Materials and methods). While this additive model reduces power relative to the correct model, our large sample size is more than able to offset the loss of power, and we find evidence d3-

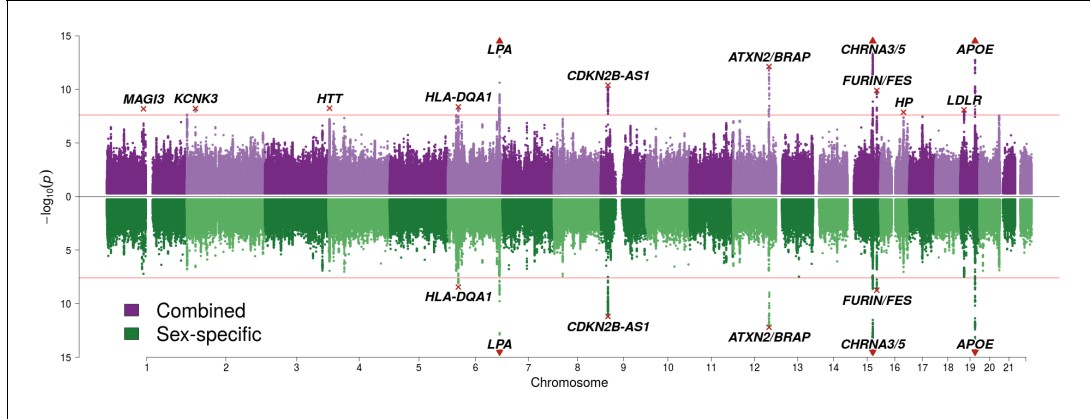

**Figure 1.** SNP associations with lifespan across both parents under the assumption of common and sex-specific effect sizes. Miami plot of genetic associations with joint parental survival. In purple are the associations under the assumption of common SNP effect sizes across sexes (CES); in green are the associations under the assumption of sex-specific effect sizes (SSE). P refers to the two-sided P values for association of allelic dosage on survival under the residualised Cox model. The red line represents our multiple testing-adjusted genome-wide significance threshold (p = 2.5 × 10$^{-8}$). Annotated are the gene, set of genes, or cytogenetic band near the index SNP, marked in red. P values have been capped at –log$_{10}$(p) = 15 to better visualise associations close to genome-wide significance. SNPs with P values beyond this cap (near *APOE*, *CHRNA3/5* and *LPA*) are represented by triangles.

DOI: https://doi.org/10.7554/eLife.39856.007

GHR does not associate with lifespan with any (recessive or additive) effect similar to that originally reported (*Figure 2*, *Figure 2—source data 1*).

## Mortality risk factor-informed GWAS (iGWAS)

We integrated 58 publicly available GWAS on mortality risk factors with our CES lifespan GWAS, creating Bayesian priors for each SNP effect based on causal effect estimates of 16 independent risk factors on lifespan. These included body mass index, blood biochemistry, CVD, type 2 diabetes, schizophrenia, multiple sclerosis, education levels, and smoking traits.

The integrated analysis reveals an additional seven genome-wide significant associations with lifespan (Bayes Factor permutation p < 2.5 × 10$^{-8}$), of which SNPs near *TMEM18*, *GBX2/ASB18*, *IGF2R*, *POM12C*, *ZC3HC1*, and *ABO* are reported at genome-wide significance for the first time (*Figure 3*; *Table 2*). A total of 82 independent SNPs associate with lifespan when allowing for a 1% false discovery rate (FDR) (*Table 2—source data 2*).

As has become increasingly common (*Pilling et al., 2017*), we attempted to replicate our genome-wide significant findings collectively, rather than individually. This is usually done by constructing polygenic risk scores from genotypic information in an independent cohort and testing for association with the trait of interest subject-by-subject. We used publicly available summary statistics on extreme longevity as an independent replication dataset (*Broer et al., 2015*; *Deelen et al., 2014*), but lacking individual data from such studies, we calculated the collective effect of lifespan SNPs on longevity using the same method as inverse-variance meta-analysis two-sample Mendelian randomisation (MR) using summary statistics (*Hemani et al., 2018*), which gives equivalent results. Prior to doing this, all effects observed in the external longevity studies were converted to hazard ratios using the *APOE* variant effect size as an empirical conversion factor, to allow the longevity studies to be meta-analysed despite their different study designs (and to be adjusted for sample overlap; see Materials and methods).

Although the focus is on collective replication, our method has the advantage of transparency at an individual variant level, which is of particular importance for researchers seeking to follow-up individual loci. Remarkably, all lead lifespan variants show directional consistency with the independent longevity sample, and 4 SNPs or close proxies (r$^2$ > 0.8) reach nominal replication (p < 0.05, one-sided test) (*Figure 4—source data 1*). Of these, SNPs near *ABO*, *ZC3HC1*, and *IGF2R* are replicated for the first time, and thus appear to affect overall survival and survival to extreme age. The overall ratio of replication effect sizes to discovery effect sizes – excluding *APOE* – is 0.42 (95% CI 0.23–

**Table 1.** Twelve genome-wide significant associations with lifespan using UK Biobank and LifeGen.
Parental phenotypes from UK Biobank and LifeGen meta-analysis, described in *Table 1—source data 1*, were tested for association with subject genotype. See *Table 1—source data 2* for LD Score regression intercept of each cohort separately and combined. Displayed here are loci associating with lifespan at genome-wide significance ($p < 2.5 \times 10^{-8}$). At or near – Gene, set of genes, or cytogenetic band nearest to the index SNP; rsID – The index SNP with the lowest P value in the standard or sex-specific effect (SSE) analysis. Chr – Chromosome; Position – Base-pair position on chromosome (GRCh37); A1 – the effect allele, increasing lifespan; Freq1 – Frequency of the A1 allele; Years1 – Years of life gained for carrying one copy of the A1 allele; SE – Standard Error; P – the P value for the Wald test of association between imputed dosage and cox model residual; Disease – Category of disease for known associations with SNP or close proxies ($r^2 > 0.6$), see *Table 1—source data 3* for details and references. Despite the well-known function of the *HTT* gene in Huntington's disease, SNPs within the identified locus near this gene have not been associated with the disease at genome-wide significance.

| At or near | rsID | Chr | Position | A1 | Freq1 | Years1 | SE | P | SSE P | Disease |
|---|---|---|---|---|---|---|---|---|---|---|
| MAGI3 | rs1230666 | 1 | 114173410 | G | 0.85 | 0.3224 | 0.0555 | 6.4E-09 | 6.1E-08 | Autoimmune |
| KCNK3 | rs1275922 | 2 | 26932887 | G | 0.74 | 0.2579 | 0.0443 | 6.0E-09 | 2.7E-07 | Cardiometabolic |
| HTT | rs61348208 | 4 | 3089564 | T | 0.39 | 0.2299 | 0.0395 | 5.8E-09 | 1.2E-07 | - |
| HLA-DQA1 | rs34967069 | 6 | 32591248 | T | 0.07 | 0.5613 | 0.0956 | 4.3E-09 | 3.6E-09 | Autoimmune |
| LPA | rs10455872 | 6 | 161010118 | A | 0.92 | 0.7639 | 0.0743 | 8.5E-25 | 3.1E-24 | Cardiometabolic |
| CDKN2B-AS1 | rs1556516 | 9 | 22100176 | G | 0.50 | 0.2510 | 0.0386 | 7.5E-11 | 6.4E-12 | Cardiometabolic |
| ATXN2/BRAP | rs11065979 | 12 | 112059557 | C | 0.56 | 0.2798 | 0.0393 | 1.0E-12 | 6.2E-13 | Autoimmune/ Cardiometabolic |
| CHRNA3/5 | rs8042849 | 15 | 78817929 | T | 0.65 | 0.4368 | 0.0410 | 1.6E-26 | 1.9E-30 | Smoking-related |
| FURIN/FES | rs6224 | 15 | 91423543 | G | 0.52 | 0.2507 | 0.0390 | 1.3E-10 | 1.8E-09 | Cardiometabolic |
| HP | rs12924886 | 16 | 72075593 | A | 0.80 | 0.2798 | 0.0493 | 1.4E-08 | 9.1E-08 | Cardiometabolic |
| LDLR | rs142158911 | 19 | 11190534 | A | 0.12 | 0.3550 | 0.0616 | 8.1E-09 | 3.3E-08 | Cardiometabolic |
| APOE | rs429358 | 19 | 45411941 | T | 0.85 | 1.0561 | 0.0546 | 3.1E-83 | 1.8E-85 | Cardiometabolic/ Neuropsychiatric |

DOI: https://doi.org/10.7554/eLife.39856.003

The following source data is available for Table 1:
Source data 1. Descriptive statistics of the cohorts and lives analysed.
Summary statistics for the 1,012,240 parental lifespans passing phenotypic QC (most notably, parent age > 40). In practice, fewer lives than these were analysed for some SNPs, as a SNP may not have passed QC in all cohorts (in particular LifeGen MAF > 1%). Ancestries in UK Biobank are self-declared, except in the case of Gen. British. Gen. British – Participants identified as genomically British by UK Biobank, based on their genomic profile. LifeGen – A consortium of 26 population cohorts of European Ancestry, with UK Biobank lives removed.
DOI: https://doi.org/10.7554/eLife.39856.004
Source data 2. LD-score regression intercepts for GWAS results.
Regression intercepts (standard error) of the GWAS summary statistics as calculated by LD-score regression, using LD scores from on average 457,407 SNPs from the UK Biobank array. CES – Results under the assumption of common effect sizes across sexes, SSE – Results allowing for sex-specific effects.
DOI: https://doi.org/10.7554/eLife.39856.005
Source data 3. Known associations with genome-wide significant lifespan loci.
Genome-wide significant associations from the GWAS catalog and PhenoScanner are reported for the lead SNP and proxies ($r^2 > 0.6$). Similar associations have been grouped, keeping the most significant association and the shortest trait name (Trait). At or near – Gene or cluster of genes in close proximity to lead SNP; A1 – the effect allele, increasing lifespan; A0 – the reference allele. Freq1- Frequency of the A1 allele in the original study, or if missing, averaged from all associations; Beta1 – the reported effect on the trait for carrying one copy of the A1 allele; SE – Standard Error; P – P value; Disease – the type of lifespan-shortening diseases linked to the trait, or 'other' if the link is unclear or multiple disease links exist.
DOI: https://doi.org/10.7554/eLife.39856.006

0.61; $p = 1.35 \times 10^{-5}$). The fact this ratio is significantly greater than zero indicates most lifespan SNPs are indeed longevity SNPs. However, the fact most SNPs have a ratio smaller than one indicates they may affect early mortality more than survival to extreme age, relative to *APOE* (which itself has a greater effect on late-life mortality than early mortality) (*Figure 4*).

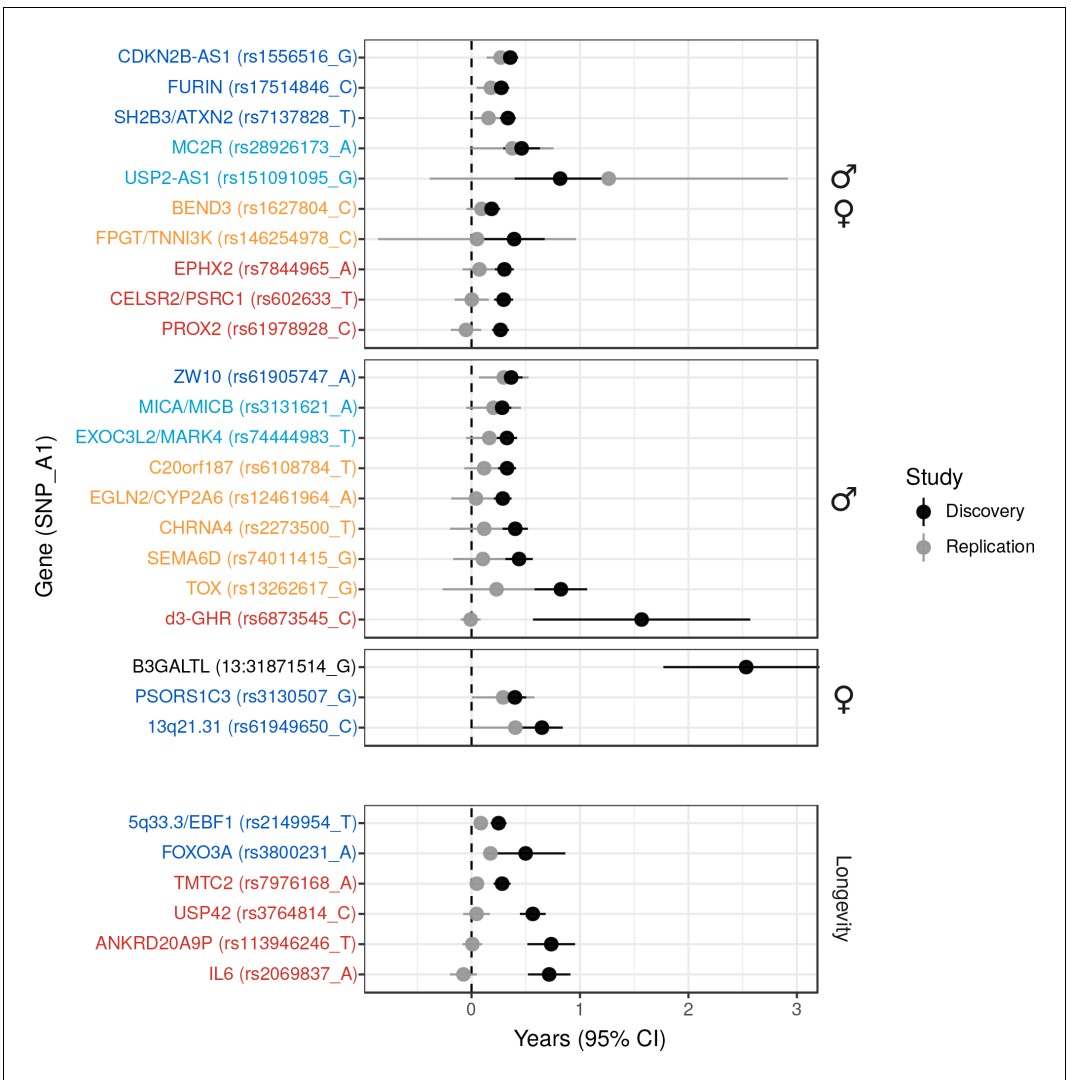

**Figure 2.** Validation of SNPs identified in other studies using independent samples of European descent. Discovery – Candidate SNPs or proxies ($r^2 > 0.95$) associated with lifespan (top panels, stratified by sex) and longevity (bottom panel) by previous studies (*Zeng et al., 2016*; *Pilling et al., 2017*; *Deelen et al., 2014*; *Flachsbart et al., 2009*; *Sebastiani et al., 2017*; *Ben-Avraham et al., 2017*). Effect sizes have been rescaled to years of life to make direct comparisons between studies (see Materials and methods and *Figure 2—figure supplement 1*). Replication – Independent samples, either the LifeGen meta-analysis to replicate *Pilling et al. (2017)*, or the full dataset including UK Biobank. Gene names are as reported by discovery and have been coloured based on overlap between confidence intervals (CIs) of effect estimates. Dark blue – Nominal replication ($p < 0.05$, one-sided test). Light blue – CIs overlap ($P_{het} > 0.05$) and cover zero, but replication estimate is closer to discovery than zero. Yellow – CIs overlap ($P_{het} > 0.05$) and cover zero, and replication estimate is closer to zero than discovery. Red – CIs do not overlap ($P_{het} < 0.05$) and replication estimate covers zero. Black – no replication data.

DOI: https://doi.org/10.7554/eLife.39856.008

The following source data and figure supplement are available for figure 2:

**Source data 1.** Eight candidate lifespan regions replicate nominally ($p < 0.05$) in LifeGen or our full sample.
DOI: https://doi.org/10.7554/eLife.39856.010

**Figure supplement 1.** Concordance between inferred effect sizes from *Pilling et al. (2017)* and our estimated effect sizes in a largely overlapping UK Biobank sample.
DOI: https://doi.org/10.7554/eLife.39856.009

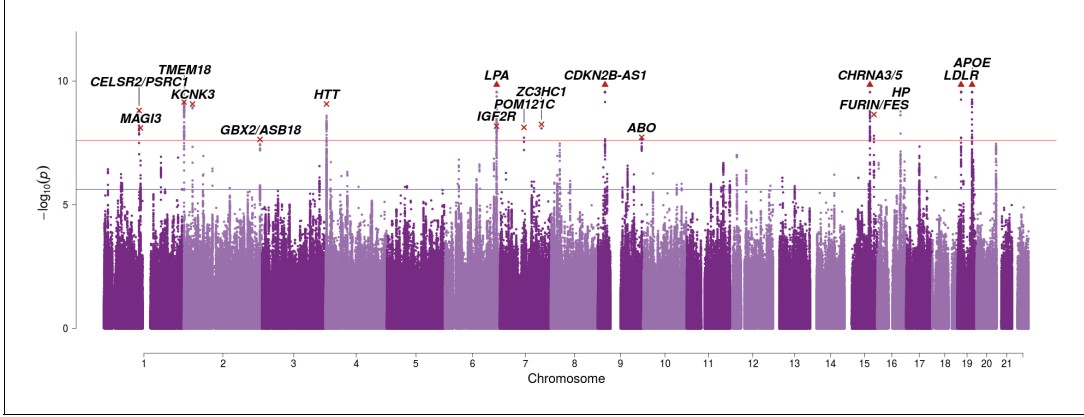

**Figure 3.** SNP associations with lifespan across both parents when taking into account prior information on mortality risk factors. Bayesian iGWAS was performed using observed associations from the lifespan GWAS and priors based on 16 traits selected by an AIC-based stepwise model. As the P values were assigned empirically using a permutation approach, the minimum P value is limited by the number of permutations; SNPs reaching this limit are represented by triangles. Annotated are the gene, cluster of genes, or cytogenetic band in close proximity to the top SNP. The red line represents the genome-wide significance threshold (p = $2.5 \times 10^{-8}$). The blue line represents the 1% FDR threshold. *Figure 3—figure supplement 1* shows the associations of each genome-wide significant SNP with the 16 risk factors.

DOI: https://doi.org/10.7554/eLife.39856.011

The following figure supplement is available for figure 3:

**Figure supplement 1.** Heat map of the effect of genome-wide significant iGWAS SNPs on the mortality risk factors.

DOI: https://doi.org/10.7554/eLife.39856.012

## Sex- and age-specific effects

We stratified our UK Biobank sample (for which we had individual level data) by sex and age bands to identify sex- and age-specific effects for survival SNPs discovered and/or replicated in this study. Although power was limited, as we sought contrasts in small effect sizes, we find 5 SNPs with differential effects on lifespan when stratified (FDR 5% across the 24 variants considered).

The effect of the *APOE* variant increases with age: the ε4 log hazard ratio on individuals older than 70 years is around 3 times greater than those between ages 40–70. In contrast, the effect of lead variants near *CHRNA3/5*, *CDKN2B-AS1*, and *ABO* tends to decline after age 60, at least when expressed as hazard ratios (*Figure 5A*).

Independent of age, lead variants near *APOE* and *PSORS1C3* also show an effect (lnHR) of 0.036; 0.038 greater in women (95% CI 0.013–0.059; 0.019–0.056, respectively), compared to men (*Figure 5B*). Notably, the SNP near *ZW10*, which was identified by *Pilling et al. (2017)* in fathers, and which replicated in LifeGen fathers, may affect men and women equally (95% CI years gained per effect allele, men 0.17–0.42, women 0.04–0.31), as measured in our meta-analysis of UK Biobank and LifeGen.

## Causal genes and methylation sites

We used SMR-HEIDI to look for causal effects of gene expression or changes in methylation on lifespan within the 24 loci discovered or replicated in our study. Using blood eQTL summary statistics from two studies (*Westra et al., 2013*; *Lloyd-Jones et al., 2017*), we suggest causal roles for expression of *PSRC1*, *SESN1*, *SH2B3*, *PSMA4*, *FURIN*, *FES*, and *KANK2* at 5% FDR (*Supplementary file 1*). GTEx tissue-wide expression data suggests further roles for 16 genes across 24 tissues, especially *FES* (nine tissues), *PMS2P3* (six tissues) and *PSORS1C1* (four tissues). Methylation data reveals roles for 44 CpG sites near nine loci, especially near the *PSORS1C3* locus (21 sites), *APOE* locus (nine sites), and *HLA-DQA1* locus (four sites) (*Supplementary file 2*).

We next used SOJO to perform conditional analysis on the same loci to find additional independent variants associated with lifespan. We find substantial allelic heterogeneity in several association intervals and identify an additional 335 variants, which increase out-of-sample explained variance from 0.095% to 0.169% (78% increase). *CELSR2/PSRC1*, *KCNK3*, *HLA-DQA1*, *LPA*, *ZW10*, *FURIN/*

**Table 2.** Bayesian GWAS using mortality risk factors reveals seven additional genome-wide significant variants.

At or near – Gene or set of genes nearest to the index SNP; rsID – The index SNP with the lowest P value in the risk factor-informed analysis. Chr – Chromosome; Position – Base-pair position on chromosome (GRCh37); A1 – the effect allele, increasing lifespan; Freq1 – Frequency of the A1 allele; Years1 – Years of life gained for carrying one copy of the A1 allele; SE – Standard Error; CES P – the P value for the Wald test of association between imputed dosage and cox model residual, under the assumption of common effects between sexes. Risk – mortality risk factors associated with the variant (p < $3.81 \times 10^{-5}$, accounting for 82 independent SNPs and 16 independent factors). BF P – Empirical P value derived from permuting Bayes Factors. See *Table 2—source data 1* for the causal estimate of each risk factor. See *Table 2—source data 2* for all SNPs significant at FDR < 1%.

| At or near | rsID | Chr | Position | A1 | Freq1 | Years1 | SE | CES P | Risk | BF P |
|---|---|---|---|---|---|---|---|---|---|---|
| CELSR2/PSRC1 | rs4970836 | 1 | 109821797 | G | 0.23 | 0.2234 | 0.0463 | 1.4E-06 | LDL HDL CAD | 1.6E-09 |
| TMEM18 | rs6744653 | 2 | 628524 | A | 0.17 | 0.2772 | 0.0511 | 5.8E-08 | BMI | 7.0E-10 |
| GBX2/ASB18 | rs10211471 | 2 | 237081854 | C | 0.80 | 0.2401 | 0.0493 | 1.1E-06 | Education | 2.3E-08 |
| IGF2R | rs111333005 | 6 | 160487196 | G | 0.98 | 0.8665 | 0.1577 | 3.9E-08 | LDL CAD | 6.6E-09 |
| POM121C | rs113160991 | 7 | 75094329 | G | 0.78 | 0.2541 | 0.0495 | 2.8E-07 | BMI Insulin | 7.5E-09 |
| ZC3HC1 | rs56179563 | 7 | 129685597 | A | 0.39 | 0.2107 | 0.0406 | 2.1E-07 | CAD | 5.6E-09 |
| ABO | rs2519093 | 9 | 136141870 | C | 0.81 | 0.2244 | 0.0497 | 6.3E-06 | LDL CAD | 1.9E-08 |

DOI: https://doi.org/10.7554/eLife.39856.013

The following source data is available for Table 2:

Source data 1. Bayesian GWAS - Multivariate effect estimates for the 16 traits chosen by the AIC based stepwise model selection.

The multivariate MR identified 16 traits (58 tested, see *McDaid et al., 2017* for an exhaustive list) with significant causal effect on lifespan and used the effect estimates to create the prior assumption of the expected effect size of each variant on lifespan, in the (Bayesian) iGWAS. Effect Estimate – the estimated effect of standardized trait on standardized lifespan, in multivariate model. SE – the standard error of the estimated effect, in multivariate model. P – the P value (two sided) from MR, for testing association between standardized trait and standardized lifespan, in multivariate model.

DOI: https://doi.org/10.7554/eLife.39856.014

Source data 2. 82 SNPs significantly associated with lifespan at 1% FDR and the SNP's associations with risk factors.

Bayesian iGWAS was performed using observed association results from CES GWAS and priors from 16 risk factors selected by AIC based stepwise model selection. Bayes Factors were calculated to compare effect estimates observed in the conventional GWAS to the prior effect computed. Empirical P values were assigned using a permutation approach and further corrected for multiple testing using Benjamini-Hochberg correction. Chr – Chromosome, Position – Base-pair position on chromosome (GRCh37), A1 – Effect Allele, Freq1 – Frequency of the A1 allele (from conventional GWAS), Beta1 (from conventional GWAS), SE – Standard Error of Beta1, Years – Years of lifespan gained for carrying one copy of the A1 allele (from conventional GWAS), P – P value (from conventional GWAS), PriorEffect – Prior effect estimate calculated from the summary statistics data for the 16 risk factors identified, PriorSE – Standard Error of the prior effect estimate, LogBF – Log of the observed Bayes Factor, P_BF – Empirical P value from a permutation approach for the log Bayes Factor. Final columns show the P value of each SNP in the studies used to calculate the prior, if the P value is significant after Bonferroni multiple testing correction (p < $3.81 \times 10^{-5}$, 82*16 tests) the cell is shaded green. Counts of these significant associations by SNP/trait are shown in the final column/row.

DOI: https://doi.org/10.7554/eLife.39856.015

---

*FES*, and *APOE* are amongst the most heterogeneous loci with at least 25 variants per locus showing independent effects (*Supplementary file 3*).

## Disease and lifespan

We next sought to understand the link between our lifespan variants and disease. We looked up known associations with our top hits and proxies ($r^2 > 0.6$) in the GWAS catalog (*MacArthur et al., 2017*) and PhenoScanner (*Staley et al., 2016*), excluding loci identified in iGWAS as these used disease associations to build the effect priors. We also excluded trait associations discovered solely in UK Biobank, as the overlap with our sample could result in spurious association due to correlations between morbidity and mortality. Under these restrictions, we find alleles which increase lifespan associate with a reduction in cardiometabolic, autoimmune, smoking-related, and neuropsychiatric disease and their disease risk factors (*Table 1*, *Table 1—source data 3*). None of the loci show any association with cancer other than lung cancer.

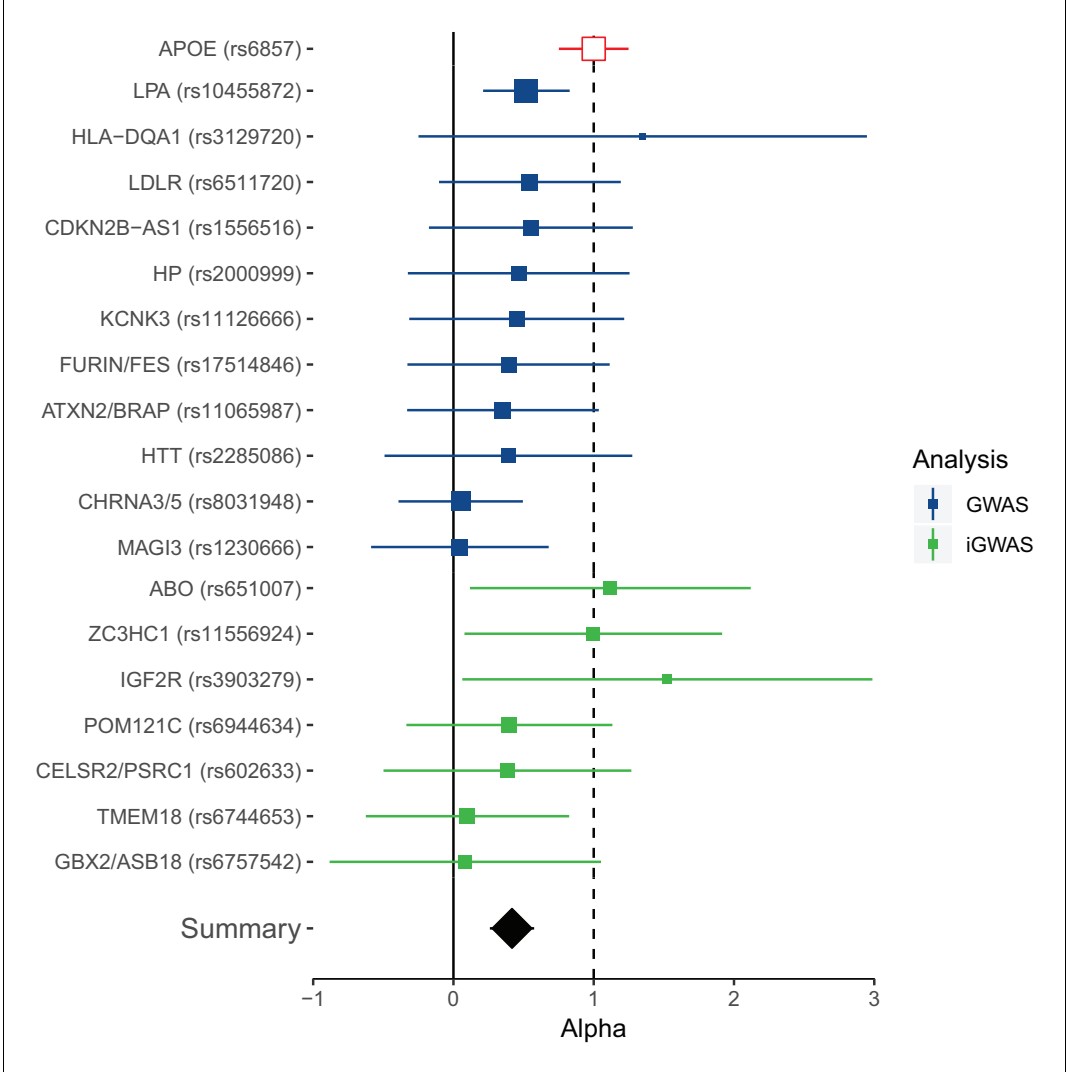

**Figure 4.** Collective replication of individual lifespan SNPs using GWAMAs for extreme long-livedness shows directional consistency in all cases. Forest plot of effect size ratios between genome-wide significant lifespan variants from our study and external longevity studies (*Broer et al., 2015*; *Deelen et al., 2014*), having converted longevity effect sizes to our scale using APOE as benchmark (see Materials and methods and *Figure 4—source data 1*). Alpha – ratio of replication to discovery effect sizes on the common scale and 95% CI (reflecting uncertainty in the numerator and denominator; P values are for one-sided test). A true (rather than estimated) ratio of 1 indicates the relationship between SNP effect on lifetime hazard and extreme longevity is the same as that of APOE, while a ratio of zero suggests no effect on longevity. A true ratio between 0 and 1 suggests a stronger effect on lifetime hazard than longevity relative to APOE. SNPs overlapping both 0 and 1 are individually underpowered. The inverse variance meta-analysis of alpha over all SNPs, excluding *APOE*, is 0.42 (95% 0.23 to 0.61; p = $1.35 \times 10^{-5}$) for $H_0$ alpha = 0.

DOI: https://doi.org/10.7554/eLife.39856.016

The following source data is available for figure 4:

**Source data 1.** Replication of lead SNPs associating with lifespan using published longevity GWAS.
DOI: https://doi.org/10.7554/eLife.39856.017

We then looked up associations of the 81 iGWAS SNPs (1% FDR) with the risk factor GWAMAs used to inform the prior. While associations are *a priori* limited to the risk factors included in the iGWAS, the pattern of association is still of interest. We find loci show strong clustering in either blood lipids or CVD, show moderate clustering of metabolic and neurological traits, and show weak

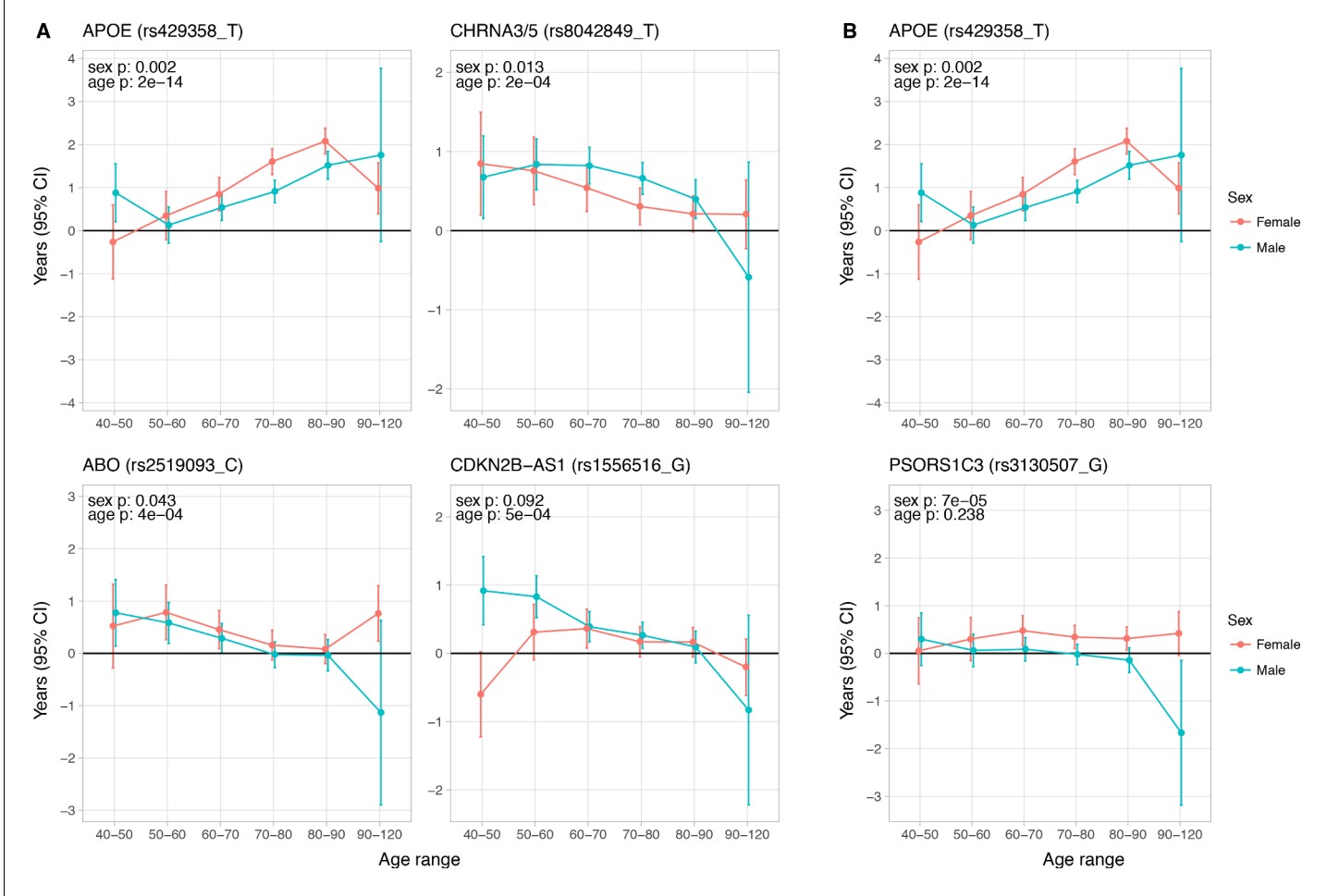

**Figure 5.** Age and sex specific effects on parent survival for 5 variants showing 5% FDR age- or sex-specificity of effect size from 23 lifespan-increasing variants. (**A**) Variants showing age-specific effects; (**B**) Variants showing sex-specific effects. Panel titles show the gene, cluster of genes, or cytogenetic band in close proximity to the index lifespan variant, with this variant and lifespan-increasing allele in parentheses. Beta – $\log_e$(protection ratio) for 1 copy of effect allele in self in the age band (i.e. 2 x observed due to 50% kinship). Note the varying scale of y-axis across panels. Age range: the range of ages over which beta was estimated. Sex p – nominal P value for association of effect size with sex. Age p – nominal P value for association of effect size with age.

DOI: https://doi.org/10.7554/eLife.39856.018

The following source data is available for figure 5:

**Source data 1.** Sex and age stratified effects on survival for 24 lifespan increasing variants.

DOI: https://doi.org/10.7554/eLife.39856.019

**Source data 2.** Effect sizes of sex and age moderators within fixed-effects with moderators' model of longevity alleles for 24 SNPs.

DOI: https://doi.org/10.7554/eLife.39856.020

but highly pleiotropic clustering amongst most of the remaining traits (see *Figure 3—figure supplement 1* for clustering of genome-wide significant SNPs).

In order to study the relative contribution of diseases to lifespan, we approached the question from the other end and looked up known associations for disease categories (CVD, type 2 diabetes, neurological disease, smoking-related traits, and cancers) in large numbers (>20 associations in each category) from the GWAS catalog (*MacArthur et al., 2017*) and used our GWAS to see if the disease loci associate with lifespan. Our measure was lifespan variance explained (LVE, years$^2$ [*Ljungquist et al., 1998*]) by the locus, which balances effect size against frequency, and is proportional to selection response and the GWAS test statistic and thus monotonic for risk of false positive lifespan associations. Taking each independent disease variant, we ordered them by LVE, excluding any secondary disease where the locus was pleiotropic.

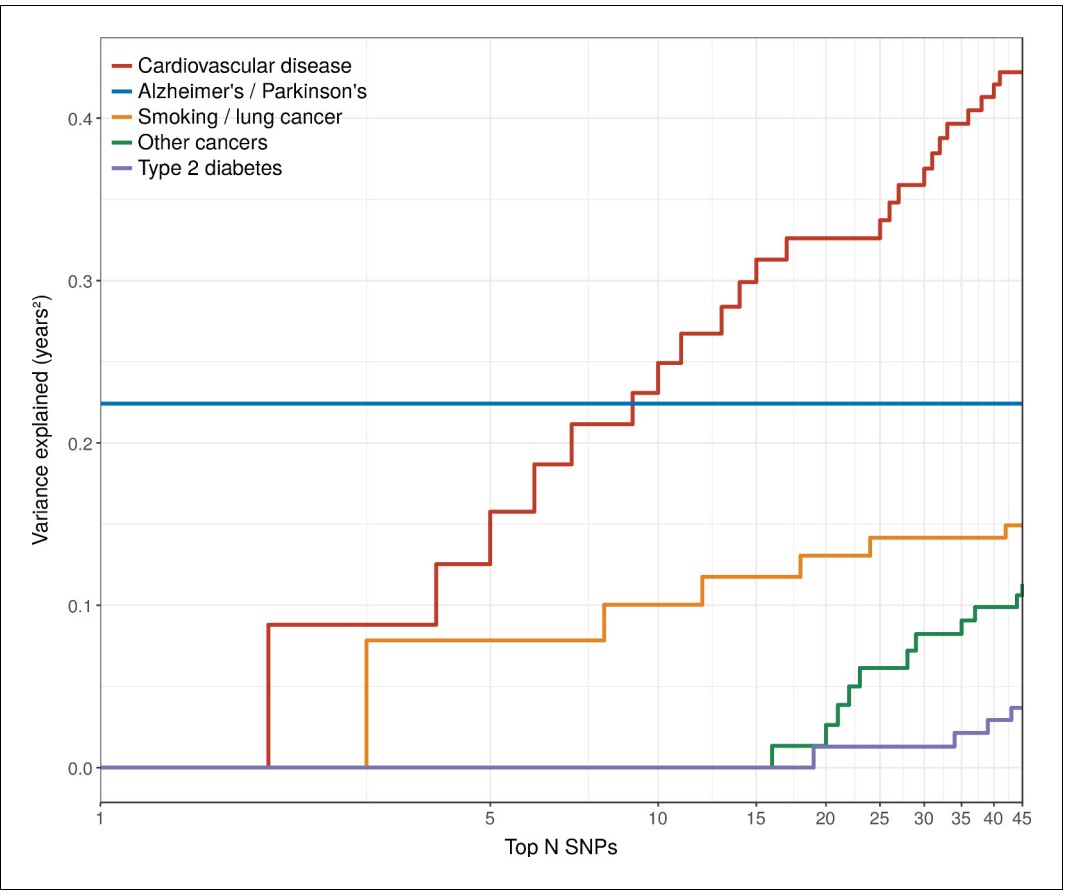

**Figure 6.** Disease loci explaining the most lifespan variance are protective for neurological disease, cardiovascular disease, and lung cancer. SNPs reported as genome-wide significant for disease in European population studies, ordered by their lifespan variance explained (LVE), show the cumulative effect of disease SNPs on variation in lifespan. An FDR cut-off of 1.55% is applied simultaneously across all diseases, allowing for one false positive association with lifespan among the 45 independent loci. Note the log scale on the X axis. Cardiovascular disease – SNPs associated with cardiovascular disease or myocardial infarction. Alzheimer's/Parkinson's – SNPs associated with Alzheimer's disease or Parkinson's disease. Smoking/lung cancer – SNPs associated with smoking behaviour, chronic obstructive pulmonary disease and lung adenocarcinomas. Other cancers – SNPs associated with cancers other than lung cancer (see *Figure 7—source data 1* for a full list). Type 2 diabetes – SNPs associated with type 2 diabetes.

DOI: https://doi.org/10.7554/eLife.39856.021

The Alzheimer's disease locus *APOE* shows the largest LVE (0.23 years$^2$), consistent with its most frequent discovery as a lifespan SNP in GWAS (*Joshi et al., 2016*; *Pilling et al., 2017*; *Deelen et al., 2014*; *Deelen et al., 2013*). Of the 20 largest LVE SNPs, 12 and 4 associate with CVD and smoking/lung cancer, respectively, while only two associate with other cancers (near *ZW10* and *NRG1*; neither in the top 15 LVE SNPs). Cumulatively, the top 20/45 LVE SNPs explain 0.33/0.43 years$^2$ through CVD, 0.13/0.15 years$^2$ through smoking and lung cancer, and 0.03/0.11 years$^2$ through other cancers (*Figure 6*).

Strikingly, two of the three largest LVE loci for non-lung cancers (at or near *ATXN2/BRAP* and *CDKN2B-AS1*) show increased cancer protection associating with decreased lifespan (due to antagonistic pleiotropy with CVD), while the third (at or near *MAGI3*) also shows evidence of pleiotropy, having an association with CVD three times as strong as breast cancer, and in the same direction. In addition, 6 out of the 11 remaining cancer-protective loci which increase lifespan and pass FDR (near *ZW10*, *NRG1*, *C6orf106*, *HNF1A*, *C20orf187*, and *ABO*) also show significant associations with CVD but could not be tested for pleiotropy as we did not have data on the relative strength of association of every type of cancer against CVD, and thus (conservatively from the point of view of our

conclusion) remain counted as cancer SNPs (*Figure 7*, *Figure 7—source data 1*). Visual inspection also reveals an interesting pattern in the SNPs that did not pass FDR correction for affecting lifespan: cardio-protective variants associate almost exclusively with increased lifespan, while cancer-protective variants appear to associate with lifespan in either direction (grey dots often appear below the x-axis for other cancers).

Together, the disease loci included in our study with significant effects on lifespan explain 0.95 years[2], or less than 1% of the phenotypic variance of lifespan of European parents in UK Biobank (123 years[2]), and around 5% of the heritability.

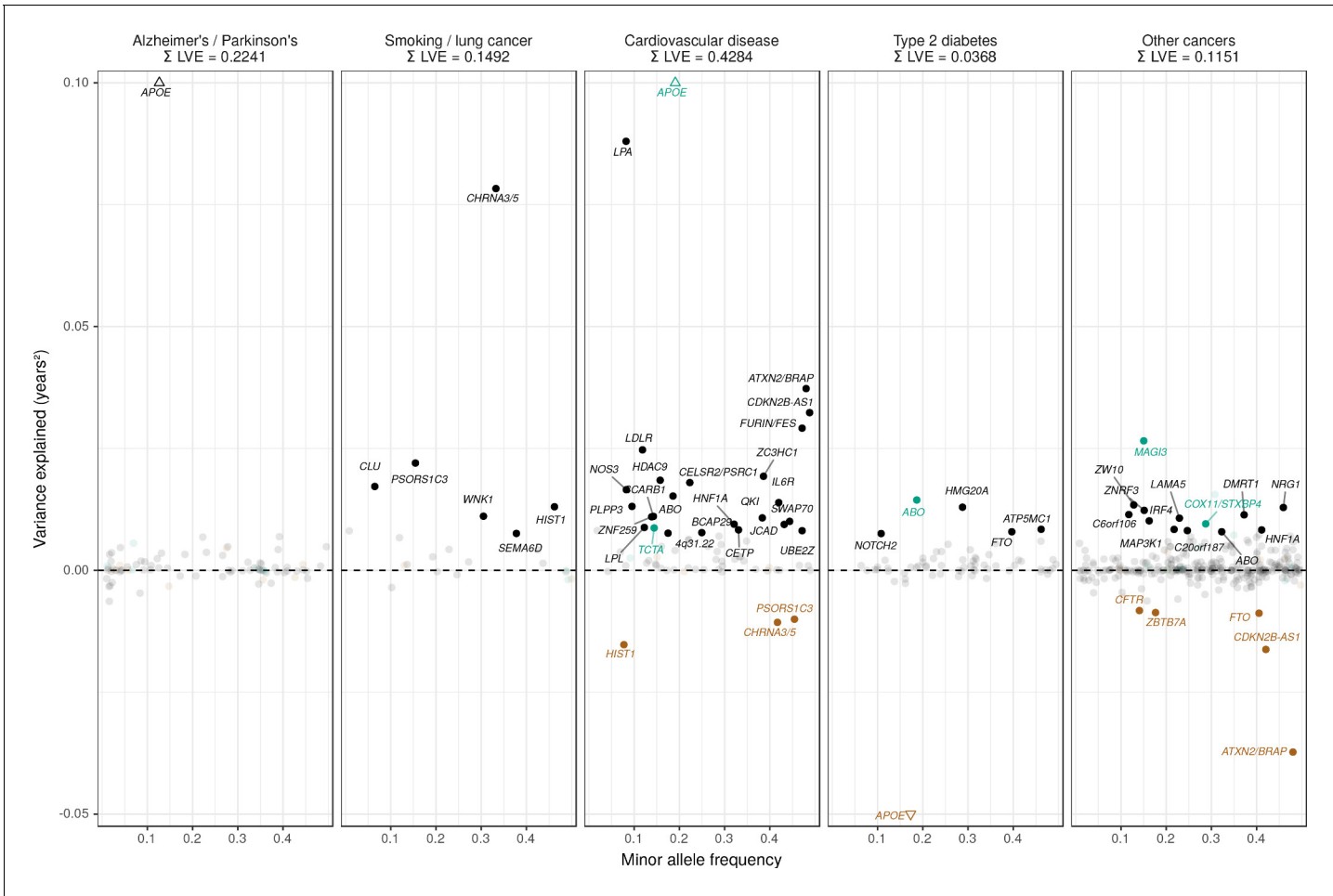

**Figure 7.** Lifespan variance explained by individual genome-wide significant disease SNPs within disease categories. Genome-wide significant disease SNPs from the GWAS catalog are plotted against the amount of lifespan variance explained (LVE), with disease-protective alleles signed positively when increasing lifespan and signed negatively when decreasing lifespan. SNPs with limited evidence of an effect on lifespan are greyed out: an FDR cut-off of 1.55% is applied simultaneously across all diseases, allowing for one false positive among all significant SNPs. Secondary pleiotropic SNPs (i.e. those associating more strongly with another one of the diseases, as assessed by PheWAS in UK Biobank) are coloured to indicate the main effect on increased lifespan seems to arise elsewhere. Of these, turquoise SNPs show one or more alternative disease associations in the same direction and at least twice as strong (double Z statistic – see Detailed Materials and methods) as the principal disease, while brown SNPs show one or more significant associations with alternative disease in the opposite direction that explains the negative association of the disease-protective SNP with lifespan. The variance explained by all SNPs in black is summed ($\sum$LVE) by disease. Annotated are the gene, cluster of genes, or cytogenetic band near the lead SNPs. The Y axis has been capped to aid legibility of SNPs with smaller LVE: SNPs near APOE pass this cap and are represented by triangles. See *Figure 7—source data 1* for the full list of disease SNP associations.

DOI: https://doi.org/10.7554/eLife.39856.022

The following source data is available for figure 7:

**Source data 1.** List of genome-wide significant disease variants, their association with disease in UK Biobank and their lifespan variance explained.
DOI: https://doi.org/10.7554/eLife.39856.023

## Cell type and pathway enrichment

We used stratified LD-score regression to assess whether cell type-specific regions of the genome are enriched for lifespan variants. As this method derives its power from SNP heritability, we limited the analysis to genomically British individuals in UK Biobank, which showed the lowest heterogeneity and the highest SNP heritability. At an FDR < 5%, we find enrichment in SNP heritability in five categories: two histone and two chromatin marks linked to male and female foetal brain cells, and one histone mark linked to the dorsolateral prefrontal cortex (DLPC) of the brain. Despite testing other cell types, such as heart, liver, and immune cells, no other categories are statistically significant after multiple testing correction (**Supplementary file 4**).

We also determined which biological pathways could explain the associations between our genetic variants and lifespan using three different methods, VEGAS, PASCAL, and DEPICT. VEGAS highlights 33 gene sets at an FDR < 5%, but neither PASCAL nor DEPICT (with SNP thresholds at p $< 5 \times 10^{-8}$ and p $< 1 \times 10^{-5}$) identify any gene sets passing multiple testing correction. The 33 gene sets highlighted by VEGAS are principally for blood lipid metabolism (21), with the majority involving lipoproteins (14) or homeostasis (4). Other noteworthy gene sets are neurological structure and function (5) and vesicle-mediated transport (3). Enrichment was also found for organic hydroxy compound transport, macromolecular complex remodelling, signalling events mediated by stem cell factor receptor (c-kit), and regulation of amyloid precursor protein catabolism (**Supplementary file 5**).

Finally, we performed an analysis to assess whether genes that have been shown to change their expression with age (**Peters et al., 2015**) are likely to have a causal effect on lifespan itself. Starting with a set of independent SNPs affecting gene expression (eQTLs), we created categories based on whether gene expression was age-dependent and whether the SNP was associated with lifespan in our study (at varying levels of significance). We find eQTLs associated with lifespan are 1.69 to 3.39 times more likely to have age-dependent gene expression, depending on the P value threshold used to define the set of lifespan SNPs (**Supplementary file 6**).

## Out-of-sample lifespan PRS associations

We calculated polygenic risk scores (PRS) for lifespan for two subsamples of UK Biobank (Scottish individuals and a random selection of English/Welsh individuals), and one sample from the Estonian Biobank. The PRS were based on (recalculated) lifespan GWAS summary statistics that excluded these samples to ensure independence between training and testing datasets.

When including all independent markers, we find an increase of one standard deviation in PRS increases lifespan by 0.8 to 1.1 years, after doubling observed parent effect sizes to compensate for the imputation of their genotypes (see **Table 3—source data 1** for a comparison of performance of different PRS thresholds).

Correspondingly – again after doubling for parental imputation – we find a difference in median survival for the top and bottom deciles of PRS of 5.6/5.6 years for Scottish fathers/mothers, 6.4/4.8 for English and Welsh fathers/mothers and 3.0/2.8 for Estonian fathers/mothers. In the Estonian Biobank, where data is available for a wider range of subject ages (i.e. beyond median survival age) we find a contrast of 3.5/2.7 years in survival for male/female subjects, across the PRS tenth to first deciles (**Table 3**, **Figure 8**).

Finally, as we did for individual variants, we looked at the age- and sex-specific nature of the PRS on parental lifespan and then tested for associations with (self-reported) age-related diseases in subjects and their kin. We find a high PRS has a larger protective effect on lifespan for mothers than fathers in UK Biobank subsamples (p = 0.0071), and has a larger protective effect on lifespan in younger age bands (p = 0.0001) (**Figure 9**), although in both cases, it should be borne in mind that women and younger people have a lower baseline hazard, so a greater improvement in hazard ratio does not necessarily mean a larger absolute protection.

We find that overall, higher PRS scores (i.e. genetically longer life) are associated with less heart disease, diabetes, hypertension, respiratory disease and lung cancer, but increased prevalence of Alzheimer's disease, Parkinson's disease, prostate cancer and breast cancer, the last three primarily in parents. We find no association between the score and prevalence of cancer in subjects. (**Figure 10**).

**Table 3.** Polygenic scores for lifespan associate with out-of-sample parent and subject lifespans.
A polygenic risk score (PRS) was made for each subject using GWAS results that did not include the subject sets under consideration. Subject or parent survival information (age entry, age exit, age of death, if applicable) was used to test the association between polygenic risk score and survival as (a) a continuous score and (b) by dichotomising the top and bottom decile scores. Population – Population sample of test dataset, where E and W is England and Wales; Kin – Individuals tested for association with polygenic score; N – Number of lives used for analysis; Deaths – Number of deaths; Beta – Effect size per PRS standard deviation, in $\log_e$(protection ratio), doubled in parents to reflect the expected effect in cohort subjects. SE – Standard error, doubled in parents to reflect the expected error in cohort subjects; Years – Estimated years of life gained per PRS standard deviation; P – P value of two-sided test of association; Contrast age at death – difference between the median lifespan of individuals in the top and bottom deciles of the score in year of life (observed parent contrast is again doubled to account for imputation of their genotypes).

| Sample descriptives | | | | Effect of polygenic score | | | | Contrast age at death | |
|---|---|---|---|---|---|---|---|---|---|
| Population | Kin | N | Deaths | Beta | SE | Years | P | Men | Women |
| Scotland | Parents | 46,936 | 33,196 | 0.107 | 0.011 | 1.07 | 4.2E-22 | 5.6 | 5.6 |
| Scotland | Subjects | 24,059 | 941 | 0.085 | 0.033 | 0.85 | 1.0E-02 | - | - |
| E and W | Parents | 58,070 | 39,347 | 0.133 | 0.010 | 1.33 | 7.3E-39 | 6.4 | 4.8 |
| E and W | Subjects | 29,815 | 760 | 0.098 | 0.037 | 0.98 | 7.1E-03 | - | - |
| Estonia | Parents | 61,728 | 29,660 | 0.099 | 0.012 | 0.99 | 2.5E-17 | 3.0 | 2.8 |
| Estonia | Subjects | 24,800 | 2894 | 0.087 | 0.019 | 0.87 | 2.6E-06 | 3.5 | 2.7 |
| | | | | Per standard deviation | | | | Top vs. bottom 10% | |

DOI: https://doi.org/10.7554/eLife.39856.024

The following source data is available for Table 3:
Source data 1. Polygenic survival scores in independent samples are most associated when including all markers.
A polygenic risk score was made for each subject using GWAS results that did not include the subject sets under consideration. Parent survival information (age and alive/dead status) was used to test the association between survival and several polygenic risk scores with different P value thresholds. Sample – Out-of-sample subsets of UK Biobank individuals used for PGRS association. N – Number of reported parental lifespans by sample individuals. Deaths – Number of reported parental deaths by sample individuals. Threshold – Criteria for SNPs to be included in the polygenic score. Beta – $\log_e$(protection ratio) per standard deviation of polygenic score, doubled to reflect the effect of the score on offspring survival. SE – standard error of the effect estimate. Mean Years – Mean years of life gained per standard deviation in PGRS. P – P value of the effect of the polygenic score on lifespan.

DOI: https://doi.org/10.7554/eLife.39856.025

## Discussion

Applying the kin-cohort method in a GWAS and mortality risk factor iGWAS across UK Biobank and the LifeGen meta-analysis, we identified 11 novel genome-wide significant associations with lifespan and replicated six previously discovered loci. We also replicated long-standing longevity SNPs near *APOE*, *FOXO3*, and 5q33.3/*EBF1* – albeit with smaller effect sizes in the latter two cases – but found evidence of no association (at effect sizes originally published) with lifespan for more recently published longevity SNPs near *IL6*, *ANKRD20A9P*, *USP42*, and *TMTC2*. Conversely, all individual variants identified in our analyses showed directionally consistent effects in a meta-analysis of two European-ancestry studies of extreme longevity, and a test of association of a polygenic risk score of the variants was highly significant in the longevity dataset ($p < 1.5 \times 10^{-5}$).

Our findings validate the results of a previous Bayesian analysis performed on a subset (N = 116,279) of the present study's discovery sample (*McDaid et al., 2017*), which highlighted two loci which are now genome-wide significant in conventional GWAS in the present study's larger sample. iGWAS thus appears to be an effective method able to identify lifespan-associated variants in smaller samples than standard GWAS, albeit relying on known biology.

With the curious exception of a locus near *HTT* (the Huntington's disease gene), all lead SNPs are known to associate with autoimmune, cardiometabolic, neuropsychiatric, or smoking-related disease, and it is plausible these are the major pathways through which the variants affect lifespan. Whole-genome polygenic risk scores showed similar associations with disease, excluding late-onset disorders such as Alzheimer's and Parkinson's, where polygenic risk scores for extended lifespan increased risk (of survival to age at onset) of the disease.

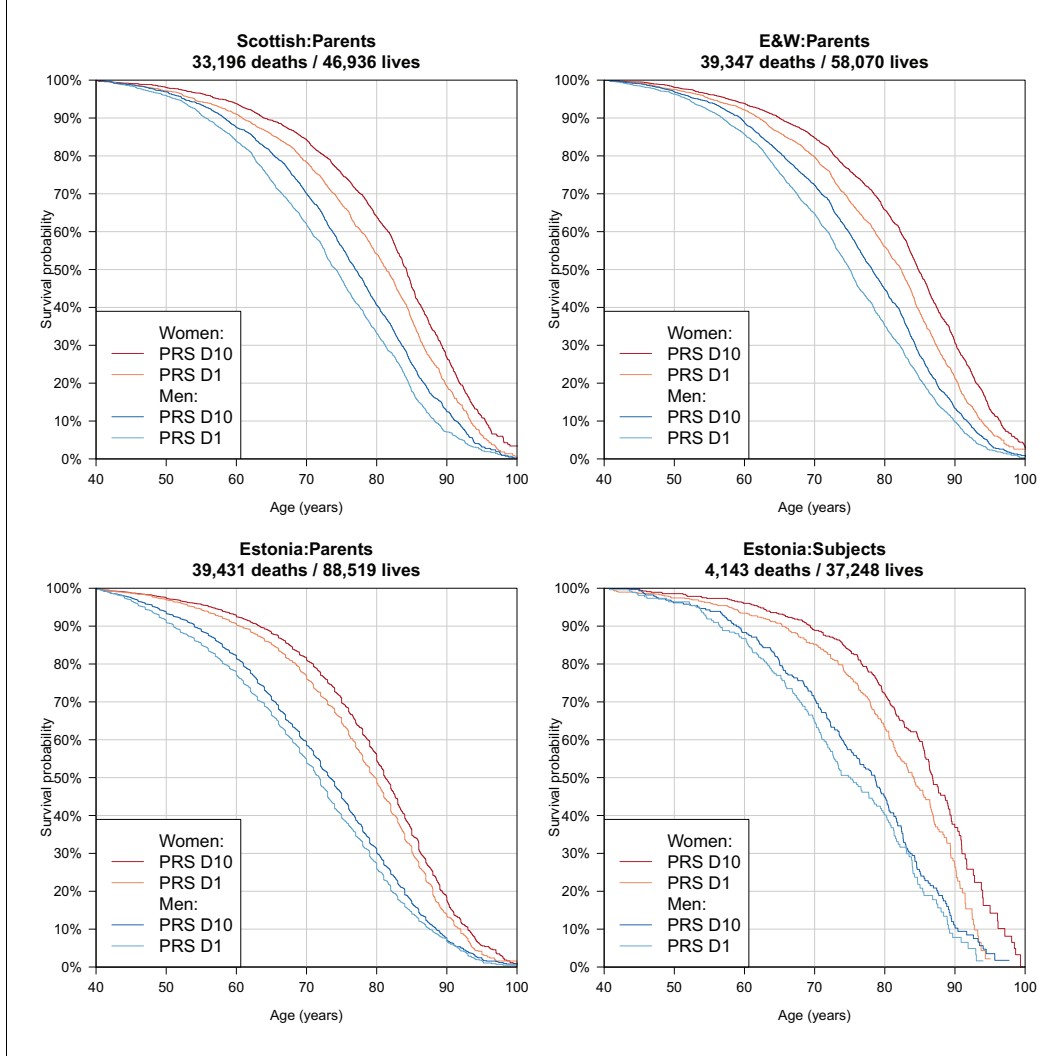

**Figure 8.** Survival curves for highest and lowest deciles of lifespan polygenic risk score. A polygenic risk score was made for each subject using GWAS results that did not include the subject sets under consideration. Subject or parent survival information (age entry, age exit, age of death (if applicable) was used to create Kaplan-Meier curves for the top and bottom deciles of score. In this figure (only) no adjustment has been made for the dilution of observed effects due to parent imputation from cohort subjects. Effect sizes in parent, if parent genotypes had been used, are expected to be twice that shown. E and W – England and Wales; PRS – polygenic risk score.
DOI: https://doi.org/10.7554/eLife.39856.026
The following figure supplement is available for figure 8:

**Figure supplement 1.** Survival Curves for highest and lowest deciles of lifespan polygenic risk score in UK Biobank subjects.
DOI: https://doi.org/10.7554/eLife.39856.027

Genetic variants affecting lifespan were enriched for pathways involving the transport, homeostasis and metabolism of lipoprotein particles, validating previous reports (*McDaid et al., 2017*). We also identified new pathways including vesicle transport, metabolism of acylglycerol and sterols, and synaptic and dendritic function. We discovered genomic regions with epigenetic marks determining cell differentiation into foetal brain and DLPC cells were enriched for genetic variants affecting lifespan. Finally, we showed that we can use our GWAS results to construct a polygenic risk score, which makes 3 to 5 year distinctions in life expectancy at birth between individuals from the score's top and bottom deciles.

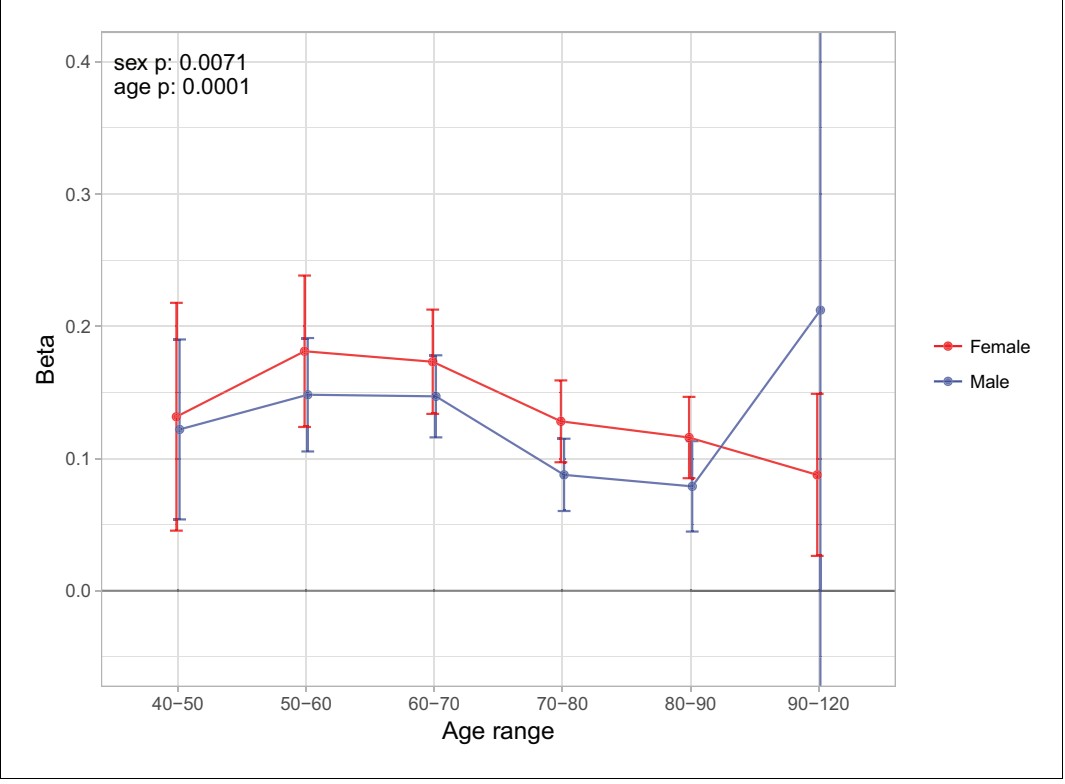

**Figure 9.** Sex and age specific effects of polygenic survival score (PRS) on parental lifespan in UK Biobank. The effect of out-of-sample PRS on parental lifespan stratified by sex and age was estimated for Scottish and English/Welsh subsamples individually (see *Figure 9—figure supplement 1*) and subsequently meta-analysed. The estimate for the PRS on father lifespan in the highest age range has very wide confidence intervals (CI) due to the limited number of fathers surviving past 90 years of age. The beta 95% CI for this estimate is –0.15 to 0.57. Beta – $\log_e$(protection ratio) for one standard deviation of PRS for increased lifespan in self in the age band (i.e. 2 x observed due to 50% kinship), bounds shown are 95% CI; Age range – the range of ages over which beta was estimated; sex p – P value for association of effect size with sex; age p – P value for association of effect size with age.

DOI: https://doi.org/10.7554/eLife.39856.028

The following source data and figure supplement are available for figure 9:

**Source data 1.** Sex and age-stratified association of polygenic score on lifespan.
DOI: https://doi.org/10.7554/eLife.39856.029

**Figure supplement 1.** Sex and age specific effects of polygenic survival score (PRS) on parental lifespan of Scottish and English/Welsh subsamples of UK Biobank.
DOI: https://doi.org/10.7554/eLife.39856.030

Despite studying over 1 million lives, our standard GWAS only identified 12 variants influencing lifespan at genome-wide significance. This contrasts with height (another highly polygenic trait) where a study of around 250,000 individuals by *Wood et al., 2014*. found 423 loci. This difference can partly be explained by the much lower heritability of lifespan (0.12; *Kaplanis et al., 2018*) (cf. 0.8 for height [*Wood et al., 2014*]), consistent with evolution having a stronger influence on the total heritability of traits more closely related to fitness and limiting effect sizes. In addition, the use of indirect genotypes (the kin-cohort method) reduces the effective sample size to 1/4 for the parent-offspring design.

When considering these limitations, we calculate our study was equal in power to a height study of only around 23,224 individuals, were lifespan to have a similar genetic architecture to height (see Materials and methods). Under this assumption, we would require a sample size of around 10 million parents (or equivalently 445,000 nonagenarian cases, with even more controls) to detect a similar number of loci as Wood *et al*. At the same time, our inability to replicate several previous borderline

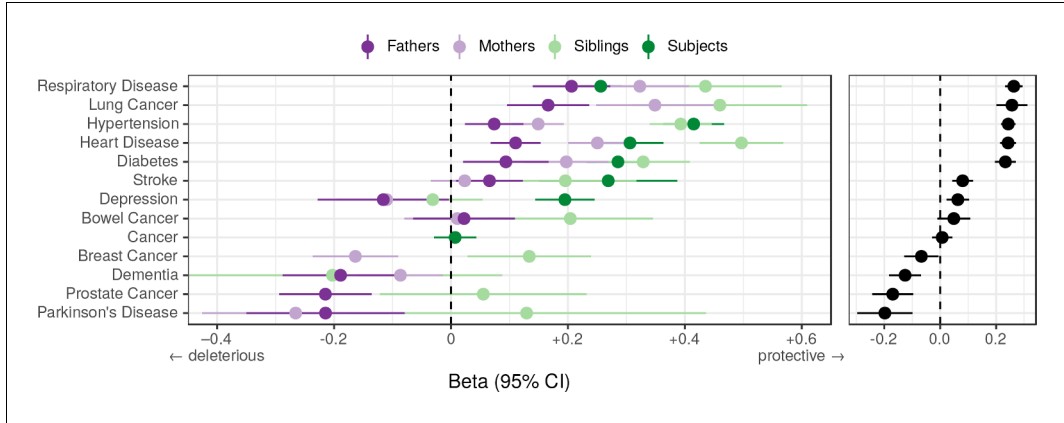

**Figure 10.** Associations between polygenic lifespan score and diseases of UK Biobank subjects and their kin. Logistic regression was performed on standardised polygenic survival score (all variants) and 21 disease traits reported by 24,059 Scottish and 29,815 English/Welsh out-of-sample individuals about themselves and their kin. For grouping of UK Biobank disease codes, see *Figure 10—source data 1*. Displayed here are inverse-variance meta-analysed estimates of the diseases for which multiple sources of data were available (i.e. parents and/or siblings; see *Figure 10—figure supplement 1* for all associations). 'Cancer' is only in subjects, whilst the specific subtypes are analysed for kin. The left panel shows disease estimates for each kin separately; the right panel shows the combined estimate, with standard errors adjusted for correlation between family members. Diseases have been ordered by magnitude of effect size (combined estimate). Beta – log odds reduction ratio of disease per standard deviation of polygenic survival score, where a negative beta indicates a deleterious effect of score on disease prevalence (lifetime so far), and positive beta indicates a protective effect on disease. Effect sizes for first degree relatives have been doubled. Cancer – Binary cancer phenotype (any cancer, yes/no).
DOI: https://doi.org/10.7554/eLife.39856.031

The following source data and figure supplement are available for figure 10:

**Source data 1.** Grouping of UK Biobank disease codes into diseases and major disease categories.
DOI: https://doi.org/10.7554/eLife.39856.032

**Source data 2.** Associations of polygenic score with diseases in UK Biobank.
DOI: https://doi.org/10.7554/eLife.39856.033

**Figure supplement 1.** Associations between polygenic survival score and diseases of individuals and their kin from Scottish and English/Welsh subsamples of UK Biobank.
DOI: https://doi.org/10.7554/eLife.39856.034

significant longevity and lifespan findings suggests research into survival in general requires substantial increases in power to robustly identify loci.

Meta-analysis of mothers and fathers, permitting common or sex-specific effect sizes, of course, doubled effective sample size, with slight attenuation to reflect the observed correlation (~10%) between father and mother traits (consistent with previous studies [*Kaplanis et al., 2018*]). This correlation indicates the presence of assortative mating on traits which correlate with lifespan (as lifespan itself is of course not observed until later), or post-pairing environmental convergence. We note that in principle, assortative mating could lead to allelic correlations at causal loci for the contributing traits, causing departures from Hardy-Weinberg equilibrium, and increasing the genotypic variance and thus power to detect association. However, in practice, at least for lifespan, the effects are too small for the effect to be material.

The association of lifespan variants with well-known, life-shortening diseases (cardiovascular, autoimmune, smoking-related diseases and lung cancer; *Mathers et al., 2018*) is not surprising, but the paucity of associations with other forms of cancer – without pleiotropic effects on CVD – is. This paucity suggests cancer deaths may often be due to (perhaps many) rarer variants or environmental exposures, although effect sizes might simply be slightly below our cut-off threshold to detect. Disappointingly, the variants and pathways we identified do not appear to underpin a generalised form of ageing independent of disease.

Our finding that lifespan genetics are enriched for lipid metabolism genes is in line with expectations, given lipid metabolites – especially cholesterol metabolites – have well-established effects on

atherosclerosis, type 2 diabetes, Alzheimer's disease, osteoporosis, and age-related cancers (*Zarrouk et al., 2014*). *Pilling et al. (2017)* implicated nicotinic acetylcholine receptor pathways in human lifespan, which we detected at nominal significance (p = $2 \times 10^{-4}$) but not at 5% FDR correction (q = 0.0556). Instead we highlighted more general synapse and dendrite pathways and identified foetal brain and DLPC cells as important in ageing. The DLPC is involved in smoking addiction (*Hayashi et al., 2013*), dietary self-control (*Lowe et al., 2014*), and is susceptible to neurodegeneration (*Morrison and Baxter, 2012*), which could explain why genetic variation for lifespan is specifically enriched in these cells, mediated through smoking-related, cardiometabolic, and neuropsychiatric disease.

Much work has been done implicating *FOXO3* as an ageing gene in model organisms (*Kenyon et al., 1993*; *Hwangbo et al., 2004*), however we found the association in humans at that locus may be driven by expression of *SESN1* (admittedly a finding restricted to peripheral blood tissue). *SESN1* is a gene connected to the *FOXO3* promoter via chromatin interactions and is involved in the response to reactive oxygen species and mTORC1 inhibition (*Donlon et al., 2017*). While fine-mapping studies have specifically found genetic variation within the locus causes differential expression of *FOXO3* itself (*Flachsbart et al., 2017*; *Grossi et al., 2018*), this does not rule out the effect of co-expression of *SESN1*. More powered tissue-specific expression data and experimental work on *SESN1* vs. *FOXO3* could elucidate the causal mechanism. For now, results from model organisms seem to leave the preponderance of evidence for *FOXO3*.

Our results suggest disease-associated lifespan variants reduce the chances of extreme long-livedness, but remain agnostic as to the more interesting two-part question: are there longevity variants that have little effect on lifespan in the normal range (*Sebastiani et al., 2016*), and if so, do they control underlying ageing processes? We note, the genetic overlap between lifespan and extreme long-livedness is high (0.73), but not complete (*McDaid et al., 2017*). Regardless of this, only a small part of the heritability of both lifespan and longevity has thus far been explained by GWAS. It thus remains plausible that an enlarged long-livedness or lifespan study will find variants controlling the rate of ageing and associated pathways. Curiously, we find little evidence of SNPs of large deleterious effect on lifespan acting with antagonistic pleiotropy on other fitness and developmental component traits, despite long-standing theoretical suggestions to the contrary (*Williams, 1957*). However, we did not examine mortality before the age of 40, or mortality of individuals without offspring (by definition as we were examining parental lifespans), which may exhibit this phenomenon. For the time being, our findings that the improved polygenic risk score for lifespan was associated with an increased prevalence of Alzheimer's disease, Parkinson's disease, and prostate and breast cancer, means we appear to be predominantly measuring a propensity for longer life through avoidance of early disease-induced mortality, rather than healthy ageing or fertility costs.

Whilst it has previously been shown that transcriptomic age calculated based on age-related genes is meaningful in the sense that its deviation from the chronological age is associated with biological features linked to ageing (*Peters et al., 2015*), the role of these genes in ageing was unclear. A gene might change expression with age because (i) it is a biological clock (higher expression tracking biological ageing, but not influencing ageing or disease); (ii) it is a response to the consequences of ageing (e.g. a protective response to CVD); (iii) it is an indicator of selection bias: if low expression is life-shortening, older people with low expression tend to be eliminated from the study, hence the average expression level of older age groups is higher. However, our results now show that the differential expression of many of the age-related genes discovered by *Peters et al., 2015* are not only biomarkers of ageing, but are also enriched for direct effects on lifespan.

There is increasing interest in polygenic risk scores, and their potential clinical utility for some diseases appears to be similar to some Mendelian mutations (albeit such monogenic tests are usually only applied in the context of family history; *Khera et al., 2018*). At first sight, the magnitude of the distinctions in our genetic lifespan score (5 years of life between top and bottom deciles, for both the parent and subject generations) are quite small compared with variability in individual lifespans. However, these distinctions are potentially material at a group level, for example, actuarially. The implied distinction in price (14%; Methods) is greater than some recently reported annuity profit margins (8.9%) (*Legal General Group PLC, 2017*). In our view, the legal and ethical frameworks (at least in the UK [*Association of British Insurers and UK Government, 2014*]) are presently underdeveloped for genome-wide scores, whether for disease or lifespan and this needs to be urgently

addressed. At the same time, although material in isolation, our lifespan associations may only have practical utility in many applications if they provide additional information than that provided by conventional clinical risk measures (e.g. the Framingham score [**D'Agostino et al., 2008**]). Such an assessment has been beyond the scope of this work, in part as such risk measures are not readily available for the parents (rather than subjects) studied.

One limitation of our study was the power reduction caused by the exclusion of relatives in our study, rather than linear mixed modelling (LMM) with a term for kinship as measured by the genomic relationship matrix (GRM) (**Pilling et al., 2017**; **Loh et al., 2015**). However, as the correct adjustment is not derivable under the kin-cohort method, we felt this was the best approach. To see that the normal adjustment is not correct, consider two siblings. The phenotypes under study are of course identical (as the parents are the same), but the expected correlation under the mixed model would only be 50% of the heritability. Simply excluding siblings, however, is not sufficient. For example, consider two offspring subjects who are first cousins descended from two full brothers. The GRM entry in this situation is 12.5% whilst the appropriate relatedness factor for the father trait is 50% and the mother 0%. Exclusion of relatives thus appears the most straightforward solution, although if a pedigree were available, not just a GRM, accurate LMM might have been feasible.

The analysis of parent lifespans has enabled us to probe mortality for a generation whose lives are mostly complete and attain increased power in a survival GWAS. However, changes in the environment (and thus the relative importance of each genetic susceptibility, for example following the smoking ban) inevitably mean we have less certainty about associations with prospective lifespans for the present generation of middle-aged people, or a different population (with perhaps different relative importance of disease or even overall heritability of lifespan). The 21% reduction in the effect size of the association between our PRS for the UK offspring generation supports this idea, although the estimated contrast in hazard ratios across the deciles was not reduced, which may be a statistical artefact or due to the different periods of life probed. The lower explanatory power of the PRS in Estonia may reflect the differing alleles and LD patterns between the UK training data and the Estonian test data, but also the different environments, in particular the sources of mortality in that country in the Soviet, and early post-Soviet era.

In conclusion, recent genomic susceptibility to death in the normal age range seems rooted in modern diseases: Alzheimer's, CVD and lung cancer; in turn arising from our modern – long-lived, obesogenic and tobacco-laden – environment, however the keys to the distinct traits of ageing and extreme longevity remain elusive. At the same time, genomic information alone can now make material distinctions at a group level in variations in expected length of life, although the limited individual accuracy of these distinctions is far from reaching genetic determinism of that most (self-) interesting of traits – your lifespan.

## Materials and methods

### Summary
GWAS

For genetically British ancestry (as identified by UK Biobank using genomic PCA) and each self-reported European ethnicity in UK Biobank (including self-declared British but not genetically British ancestry), independent association analyses were performed between unrelated subjects' genotypes (MAF > 0.005; HRC imputed SNPs only; ~9 million markers) and parent survival using age and alive/dead status in residualised Cox models, as described in **Joshi et al. (2017)**. To account for parental genotype imputation, effect sizes were doubled, yielding log hazard ratios for the allele in carriers themselves. These values were negated to obtain a measure of log protection ratio, where higher values indicate longer life. While methods exist to account for related individuals using linear mixed models, such as BOLT-LMM (**Loh et al., 2015**), these are not accurate when trying to account for relatedness between parents (See Detailed Materials and methods).

Mother and father survival information was combined in two separate ways, essentially assuming the effects were the same in men and women, or allowing for sex-specific effect sizes (SSE), with appropriate allowance for the covariance amongst the traits. For the first analysis we summed parental survival residuals; for the second analysis we used MANOVA, implemented in MultiABEL (**Shen et al., 2015**).

For LifeGen, where individual-level data was not available, parent survival summary statistics were combined using conventional fixed-effects meta-analysis, adjusted to account for the correlation between survival traits (estimated from summary-level data). For SSE, the same procedure was followed as for the UK Biobank samples, with correlation between traits again estimated from summary-level data. The GWAS statistics showed acceptable inflation, as measured by their LD-score regression intercept (<1.06, *Table 1—source data 2*).

## Candidate SNP replication

Effect sizes from longevity studies were converted to our scale using an empirical conversion factor, based on the observed relationships between longevity and hazard ratio at the most significant variant at or near *APOE*, observed in the candidate SNPs study and our data (*Joshi et al., 2017*). These studies were then meta-analysed using inverse variance weighting and standard errors were inflated to account for sample overlap (see Detailed Methods)

Estimates reported in *Pilling et al. (2017)* were based on rank-normalized Martingale residuals, unadjusted for the proportion dead, which – for individual parents – could be converted to our scale by multiplying by sqrt(c)/c, where c is the proportion dead in the original study (see Detailed Methods for derivation). Combined parent estimates were converted using the same method as the one used for longevity studies.

The deletion reported by *Ben-Avraham et al. (2017)* is perfectly tagged by a SNP that we used to assess replication. Assuming a recessive effect and parental imputation, we derived the expected additive effect to be $\hat{\beta}_C = \hat{\beta}_{CC} \frac{q^2}{q^2+2pq}$, where $\hat{\beta}_C$ is the effect we expect to observe under our additive model, $\hat{\beta}_{CC}$ is the homozygous effect reported in the original study, $q$ is the C allele frequency, and $p$ is $1 - q$ (see Detailed Materials and methods for derivation).

## iGWAS

58 GWAS on mortality risk factors were used to create Bayesian priors for the SNP effects observed in the CES study, as described in *McDaid et al. (2017)*. Mendelian randomisation was used to estimate causal effects of independent risk factors on lifespan, and these estimates were combined with the risk factor GWAS to calculate priors for each SNP. Priors were multiplied with observed Z statistics and used to generate Bayes factors. Observed Z statistics were then permuted, leading to 7.2 billion null Bayes factors (using the same priors), which were used to assess significance.

## Sex and age stratified analysis

Cox survival models, adjusting for the same covariates as the standard GWAS, were used to test SNP dosage against survival of UK Biobank genomically British fathers and mothers, separately. The analysis was split into age bands, where any parent who died at an age younger than the age band was excluded and any parent who died beyond the age band was treated as alive. Using the R package 'metafor', moderator effects of sex and age on hazard ratio could be estimated while taking into account the estimate uncertainty (see Detailed Materials and methods for formula).

## Causal genes and methylation sites

SMR-HEIDI (*Zhu et al., 2016*) tests were performed on CES statistics to implicate causal genes and methylation sites. Summary-level data from two studies on gene expression in blood (*Westra et al., 2013*; *Lloyd-Jones et al., 2017*) and data on gene expression in 48 tissues from the GTEx consortium (*Battle et al., 2017*) were tested to find causal links between gene expression and lifespan. Similarly, data from a genome-wide methylation study (*McRae et al., 2017*) was used to find causal links between CpG sites and lifespan. All results from the SMR test passing a 5% FDR threshold where the HEIDI test p>0.05 were reported.

## Conditional analysis

SOJO (*Ning et al., 2017*) was used to fine-map the genetic signals in 1 Mb regions around lead SNPs reaching genome-wide significance and candidate SNPs reaching nominal significance in our study. The analysis was based on CES statistics from UK Biobank genomically British individuals, using the LifeGen meta-analysis results to optimise the LASSO regression tuning parameters. For

each parameter, a polygenic score was built and the proportion of predictable variance from the regional polygenic score in the validation sample was calculated.

## Disease association analysis

The GWAS catalog (*MacArthur et al., 2017*) and PhenoScanner (*Staley et al., 2016*) were checked for known genome-wide significant associations with our GWAS hits and proxies (r2 >0.6) in European samples. Associations discovered in UK Biobank by *Churchhouse and Neale, 2018* were omitted from the PhenoScanner database as the findings have not been replicated, and the large sample overlap with our own study could result in false positive associations, due to phenotypic correlations between morbidity and mortality. Triallelic SNPs and associations without effect sizes were excluded before near-identical traits were grouped together, discarding all but the strongest association and keeping the shortest trait name. For example, 'Lung cancer', 'Familial lung cancer', and 'Small cell lung cancer' were grouped and renamed to 'Lung cancer'. The remaining associations were classified into disease categories based on keywords and subsequent manual curation.

## Lifespan variance explained by disease SNPs

The GWAS catalog (*MacArthur et al., 2017*) was checked for disease associations discovered in European ancestry studies, which were grouped into broad disease categories based on keywords and manual curation (see *Figure 7—source data 1* and Detailed Methods). Associations were pruned by distance (500 kb) and LD ($r^2$ <0.1), keeping the SNP most strongly associated with lifespan in the CES GWAS. Where possible this SNP was tested against diseases in UK Biobank subjects and their family to test for pleiotropy (see Detailed Matrials and methods). Significance of associations with lifespan was determined by setting an FDR threshold that allowed for one false positive among all independent SNPs tested ($q \leq 0.022$). Lifespan variance explained (LVE) was calculated as $2pqa^2$, where p and q are the frequencies of the effect and reference alleles in our lifespan GWAS, and a is the SNP effect size in years of life (*Falconer et al., 1996*).

## Cell type enrichment

Stratified LD-score regression (*Finucane et al., 2015*) was used to test for cell type-specific enrichment in lifespan heritability. As the power of this method depends on SNP heritability, standard LD-score regression (*Finucane et al., 2015*) was first used to check which of our samples (UK Biobank, LifeGen, or the combined cohort) had the highest SNP heritability. Lifespan summary statistics from UK Biobank genomically British individuals were then analysed using the procedure described in *Finucane et al., 2015*, and P values were adjusted for multiple testing using the Benjamin-Hochberg procedure.

## Pathway enrichment

VEGAS2 v2.01.17 (*Mishra and Macgregor, 2015*) was used to calculate gene scores using SNPs genotyped in UK Biobank, based on summary statistics from the full CES cohort and the default software settings. VEGAS2Pathway was then used to check for pathway enrichment using those gene scores and the default list of gene sets (*Mishra and MacGregor, 2017*).

DEPICT (*Pers et al., 2015*) was also used to map genes to lifespan loci and check for pathway enrichment in the combined cohort CES GWAS. Default analysis was run for regions with genome-wide significant (p < 5e-8) variants in the first analysis, and genome-wide suggestive (p < 1e-5) variants in the second analysis, excluding the MHC in both cases.

PASCAL (*Lamparter et al., 2016*) was used with the same summary statistics and gene sets as DEPICT, except the gene probabilities within the sets were dichotomized (Z > 3) as described in *Marouli et al., 2017*.

For each software independently, pathway enrichment was adjusted for multiple testing using the Benjamin-Hochberg procedure.

## Age-related eQTL enrichment

Combined cohort CES lifespan statistics were matched to eQTLs associated with the expression of at least one gene (p < $10^{-3}$) in a dataset from the eQTLGen Consortium (31,684 individuals) (*Võsa et al., 2018*). Data on age-related expression (*Peters et al., 2015*) allowed eQTLs to be

divided into four categories based on association with age and/or lifespan. Fisher's exact test was used check if age-related eQTLs were enriched for associations with lifespan.

## Polygenic score analysis

Polygenic risk scores (PRS) for lifespan were calculated for two subsamples of UK Biobank (24,059 Scottish individuals and a random 29,815 English/Welsh individuals), and 36,499 individuals from the Estonian Biobank, using combined cohort CES lifespan summary statistics that excluded these samples. PRSice 2.0.14.beta (*Euesden et al., 2015*) was used to construct the scores from genotyped SNPs in UK Biobank and imputed data from the Estonian Biobank, pruned by LD ($r^2$ = 0.1) and distance (250 kb). Polygenic scores were Z standardised.

Cox proportional hazard models were used to fit parental survival against polygenic score, adjusted for subject sex; assessment centre; genotyping batch and array; and 10 principal components. Parental hazard ratios were converted into subject years of life as described in the GWAS method section.

Logistic regression models were used to fit polygenic score against the same self-reported UK Biobank disease categories used for individual SNPs. Effect estimates of first-degree relatives were doubled to account for imputation of genotypes and then meta-analysed using inverse variance weighting, adjusting for trait correlations between family members.

## Postulation of equivalent sample size in height GWAS

The use of parent imputation, low trait heritability, and incomplete proportion dead all reduce the power to detect effect sizes. The equivalent sample size in a hypothetical, completely heritable trait with otherwise identical genetic architecture would be the original sample size, diluted (i.e. multiplied) by the heritability (0.122) (*Kaplanis et al., 2018*), the $r^2$ of offspring genotype on parent genotype (0.250) and the proportion dead (0.602). This gives an equivalent sample size of 18,579 from the 1,012,240 parent lifespans. We then calculated sample size for height that would have the same properties, accounting for the heritability of height (0.8) (*Wood et al., 2014*): 23,224 (i.e. 18,579/ 0.8). We next calculated the P value that would have been reported by Wood *et al*'s 250,000 sample size height GWAMA (*Wood et al., 2014*) for a SNP that was just significant in a hypothetical 23,224 sample height GWAMA: $p < 1.8 \times 10^{-72}$. Six distinct loci passed this significance threshold in Wood *et al*'s results.

With 17,893 nonagenarians, *Deelen et al. (2014)* attained a P value of $2.33 \times 10^{-26}$ at rs4420638. With 1.012 m parents we attained a P value of $1.75 \times 10^{-64}$. Other things being equal a nonagenarian sample size of 44,500 thus appears to be equally powered to one million parents.

## Sensitivity of annuity prices to age

Market annuity rates for life annuities in January 2018 written to 55, 60, 65, and 70 year olds were obtained from the sharing pensions website http://www.sharingpensions.com/annuity_rates. htm (accessed 22 January 2018) and were £4158, £4680, £5476, £6075, £7105 respectively per year for a £100,000 purchase price. The arithmetic average increase from one quinquennial age to the next is 14 percent.

## Data availability

Individual phenotypic and genetic data is available from UK Biobank upon application: https://www. ukbiobank.ac.uk/register-apply/. Phenotypes used in this work include subject age, sex, ethnicity, relatedness, genotyping batch, array, and principal components, as well as parental age and alive/ dead status. Also included are self-reported diseases of subjects and their families. A full description of our application can be found at https://www.ukbiobank.ac.uk/2015/02/dr-james-wilson-university-of-edinburgh-centre-for-population-sciences/. The results that support our findings, in particular, the GWAS summary statistics we generated for >1 million parental lifespans in this study are available at http://dx.doi.org/10.7488/ds/2463. Gene expression data used in the age-related gene analysis is being made available by the eQTLGen Consortium, see http://www.eqtlgen.org/ and *Võsa et al., 2018*. Single tissue gene expression data used in the SMR-Heidi analysis can be found on the GTEx website https://gtexportal.org/home/datasets, under GTEx_Analysis_v7_eQTL.tar.gz.

## Details

### Data sources

Our UK Biobank dataset consisted of 409,700 British individuals (determined by genomic PCA) and 48,656 European individuals of self-reported (but not genetic) British, Irish, and other White European descent. Details on genotyping marker and sample QC are described in *Bycroft et al., 2017*. Subjects completed a questionnaire which included questions on adoption status, parental age, and parental deaths. For our analysis, we excluded individuals who were adopted or otherwise unclear about their adoption status (N = 7,279), individuals who did not report their parental ages (N = 2,995), and individuals both of whose parents died before the age of 40 and which were therefore more likely due to accident or injury (N = 4,472). We further excluded one of each pair from related individuals (third degree or closer; N = 88,354) from every relative pair reported by UK Biobank, leaving 443,610 individuals for the final analysis. Although exclusion of relatives reduces sample size, we were concerned that linear mixed modelling to account for relatedness might not be fully appropriate under the kin-cohort model. Consider the parental phenotypic correlation for two full sibling subjects ($r^2 = 1$) or the maternal genetic covariance amongst two subjects who are the offspring of two brothers ($r^2 = 0$): the heritability/GRM implied covariance is incorrect for both cases (although in the sibling case, it may be correct on average). Individuals passing QC reported a total of 339,732 paternal and 351,889 maternal lifespans, ranging from 40 to 107 years of life, that is 691,621 lives in total (*Table 1—source data 1*).

Our second dataset was the publicly available summary statistics from LifeGen, a consortium of 26 population cohorts investigating genomic effects on parental lifespans (*Joshi et al., 2017*). LifeGen had included results from UK Biobank, but the UK Biobank GWAS data were removed here, giving GWAS summary statistics for 160,461 father and 160,158 mother lifespans in the form of log hazard ratios. Combined, our datasets had 1,012,240 lives.

### UK Biobank Genome-Wide association study

In each separate UK Biobank ethnicity, we carried out association analysis between genotype (MAF > 0.005; HRC imputed SNPs only; ~9 million markers) and parent age and alive/dead status, effectively analysing the effect of genotype in offspring on parent survival, given survival to at least age 40, using Cox Proportional Hazards models. The following model was assumed to hold:

$$h(x) = h_0(x)e^{\beta X + \gamma 1 Z1 + \ldots \gamma k Zk} \tag{1}$$

Where x is (parent) age, $h_0$ the baseline hazard and X the offspring genotype (coded 0,1,2), beta the $\log_e$(hazard ratio) associated with X and $Z_1$-$Z_k$ the covariates, with corresponding effect sizes $y_1$-$y_k$. The covariates were genotyping batch and array, the first forty principal components of relatedness, as provided by UK Biobank, and subject sex (but not age, as we were analysing parent age).

To facilitate practical runtimes, the Martingale residuals of the Cox model were calculated for father and mothers separately and divided by the proportion dead to give estimates of the hazard ratio (*Therneau et al., 1990*) giving a residual trait suitable for GWAS (for more details of the residual method see *Joshi et al., 2017*). Effect sizes observed under this model, for a SNP in offspring, are half that of the actual effect size in the parent carrying the variant (*Joshi et al., 2017*). Reported effect sizes (and their SE) have therefore been doubled to give the effect sizes in carriers themselves, giving an estimate of the log hazard ratios (or often, with sign reversed, log protective ratios). These estimates are suitable for meta-analysis and allow direct comparison with the log hazard ratios from LifeGen.

Analysis of association between genotype and survival across both parents was made under two contrasting assumptions and associated models, which had to adjust for the covariance amongst parent traits, preventing conventional unadjusted inverse-variance meta-analysis. Firstly, we assumed that the hazard ratio was the same for both sexes, that is a common effect size across sexes (CES). If there were no correlation amongst parents' traits, this could have been done by straightforward inverse variance meta-analysis of the single parent results. However, to account for the covariance amongst father and mother lifespans, we calculated a total parent residual, the sum of individual parent residuals, for each subject (i.e. offspring). Under the common effect assumption, the combined trait's effect size is twice that in the single parent, and the variance of the combined trait, automatically and appropriately reflects the parents' covariance, amongst the two parents, giving a

residual trait suitable for GWAS, with an effect size equal to that in a carrier of the variant, and correct standard error. Secondly, we assumed that, there might be sex-specific effect sizes (SSE) in fathers and mothers. Under the SSE assumption, individual parental GWAS were carried out, and the summary statistics results were meta-analysed using MANOVA, accounting for the correlation amongst the parent traits and the sample overlap (broadly complete), but agnostic as to whether the effect size was similar or different in each parent, giving a P value against the null hypothesis that both effect sizes are zero, but, naturally, no estimate of a single common beta. This procedure was carried out using the R package MultiABEL and summary-level data (*Shen et al., 2015*). The procedure requires an estimate of the correlation amongst the traits (in this case parent residuals), which was measured directly (r = 0.1). The procedure automatically estimates the variance of the traits from summary level data (Mother residuals $\sigma^2$ = 6.74; Father residuals $\sigma^2$ = 5.25)

For the LifeGen results, the SSE procedure to combine results was identical to UK Biobank (Mother residuals $\sigma^2$ = 14.12; Father residuals $\sigma^2$ = 18.75), except the trait correlation was derived from summary level data instead of measured directly (r = 0.1). This was done by taking the correlations in effect estimates from independent SNPs from the summary statistics of the individual parents, which equals the trait correlation, assuming full sample overlap (which is slightly conservative). Similarly, since we did not have access to individual level (residual) data, it was not possible to carry out a single total parent residual GWAS under the CES assumption. Instead we meta-analysed the single parent effect sizes using inverse variance meta-analysis, but adjusted the standard errors to reflect the correlation amongst the traits (r) as follows:

$$\mathrm{SE}(\hat{\beta}) = \mathrm{SE}_0(\hat{\beta})\sqrt{1+r}$$

Where $\mathrm{SE}_0(\hat{\beta})$ is the usual (uncorrected) inverse-variance weighted meta-analysis standard error, ignoring the correlation amongst the estimates and $\mathrm{SE}(\hat{\beta})$ is the corrected estimate used.

This is slightly conservative as

$$\mathrm{Variance}(\hat{\beta}) = \mathrm{Variance}_0(\hat{\beta})\left(1 + \frac{2rs_1s_2}{s_1^2 + s_2^2}\right) <= \mathrm{Variance}_0(\hat{\beta})(1+r) \tag{2}$$

which follows straight forwardly from $\hat{\beta} = \frac{P_1\hat{\beta}_1 + P_2\hat{\beta}_2}{P_1 + P_2}$.

Where $s_1$ and $s_2$ are the standard error of the individual estimates and $P_1$, $P_2$ their associated precisions (i.e. reciprocal of the variance). Equation (2) is always conservative, but exact if $s_1 = s_2$. In practice $s_1$ and $s_2$ were similar, as the sample sizes, allele frequencies and variance in the traits for the two parents were very similar.

As we were using unrelated populations and fitting forty principal components to control for population structure, material inflation of test statistics due to structure or relatedness was not to be expected. This was confirmed using the intercept of LD-score regression (*Bulik-Sullivan et al., 2015*) for the summary statistics as shown in *Table 1—source data 2*. We have tried to use a consistent approach to the direction of lifespan altering effects: positive implies longer life, consistent with previous studies of long-livedness (*Deelen et al., 2014*). Our base measure was thus a protection ratio, directly mirroring the cox hazard ratio. Effect sizes (betas) are typically $-\log_e$(cox hazard ratio), which we denote the $\log_e$(protection ratio). Years of life gained were estimated as 10 * log protection ratio, in accordance with a long-standing actuarial rule of thumb and recently verified (*Joshi et al., 2017*).

## Candidate SNP replication

We sought to reproduce and replicate genome-wide significant associations reported by *Pilling et al. (2017)*, who recently published a GWAS on the same UK Biobank data, but using a slightly different method. Rather than excluding relatives, Pilling *et al.* used BOLT-LMM and the genomic relationship matrix in subjects, to approximately account for covariance amongst parental phenotype. Pilling *et al.* also analysed parents separately as well as jointly, using a last survivor phenotype. Despite these factors, reproduction (obtaining the same result from almost the same data) was straightforward and consistent, once effect sizes were placed on the same scale (see below and *Figure 2—figure supplement 1*), confirming our re-scaling was correct. To try to independently replicate their results, we used the consortium, LifeGen, excluding individuals from UK Biobank.

*Pilling et al. (2017)* performed multiple parental survival GWAS in UK Biobank, identifying 14 loci using combined parent lifespan and 11 loci using individual parent lifespan. Their study design involved rank-normalising Martingale residuals before regressing against genotype, which does not give an estimate of the $\log_e$(hazard ratio) (lnHR), nor, we believe, another naturally interpretable scale of effects, as the scale is now dependent on the proportion dead. Simulations (not shown) suggested $sd \approx \sqrt{c}$ for some Martingale residual distributions, where sd is the standard deviation of the distribution and c is the proportion dead. As multiplying the untransformed Martingale residual distribution by 1/c gives an estimate of the hazard ratio (*Joshi et al., 2017*; *Therneau et al., 1990*), for individual parents, we could convert Pilling *et al.*'s effect sizes by multiplying them by sqrt(c) to return them to the Martingale residual scale (which still depends on the study structure) and then by 1/c to place them on the lnHR scale, using the proportion dead from Pilling *et al.*'s study descriptives. Further multiplication by 2 allows conversion from a subject-parent effect to an effect in self. The cumulative scale conversion allowing for all three of these effects was to multiply Pilling *et al.*'s effect sizes by 2.5863/2.2869 in mothers/fathers, respectively, placing them on a lnHR scale for effects in male/female subjects. The joint life parent phenotype does not appear to have a straightforward conversion to lnHR in self. Instead, we used an empirical estimate derived from effect sizes comparison of the APOE allele between Pilling *et al.*'s discovery sample and our own UK Biobank Gen. British sample (both parents combined), which were largely overlapping: to get from Pilling *et al.*'s effect size to $\log_e$ HR, we had to multiply their effect sizes by 1.9699 for *APOE* and used this ratio for other alleles, which should be completely valid under the proportional hazard assumption. Whilst this scheme may appear a little *ad hoc* (the use of simulation and *APOE*), it was confirmed empirically: visual inspection indeed showed hazard ratios from our own UK Biobank Gen. British sample calculations and inferred hazard ratios from Pilling were highly concordant (noting the concordance for joint life at *APOE*, which was pre-defined to be perfectly concordant by our procedure, is not, of itself, evidence) (*Figure 2—figure supplement 1*).

*Flachsbart et al. (2009)*, *Deelen et al. (2014)*, and *Broer et al. (2015)* tested extreme longevity cases (95–110 years, ≥85 years, ≥90 years, respectively) against controls (60–75 years, 65 years, deceased at 55–80 years, respectively), identifying SNPs at or near *FOXO3* and 5q33.3/*EBF1*. As done previously (*Joshi et al., 2017*), we estimated the relationship between study-specific longevity log odds ratio and log hazard ratio empirically using the most powered *APOE* variant overlapping with our own study, assuming increased odds of surviving to extreme age is due to a reduction in lifetime mortality risk. For Flachsbart *et al.* and Deelen *et al.*, we used rs4420638_G (reported log OR –0.33 (*Deelen et al., 2014*), our lnHR 0.086). Inverting the sign to give $\log_e$(protection ratio) estimates, the conversion estimate used was 3.82. For Broer *et al.*, we used rs6857_T (reported log OR –0.20 (*Broer et al., 2015*), our lnHR 0.087), with a conversion estimate of 0.43, again yielding a $\log_e$(protection ratio).

*Ben-Avraham et al. (2017)* reported a deletion in Growth Hormone Receptor exon 3 (*d3-GHR*) associated with an increase of 10 years in male lifespan when homozygous. This deletion is tagged by rs6873545_C (*McKay et al., 2007*), which is present in the UK Biobank and LifeGen population sample at a frequency of 26.9% ($q$). Considering the association is recessive and we are imputing father genotypes, we converted the reported effect size into expected years of life per allele as follows:

If the subject genotype is CT, the parent contributing the C allele has 50% chance of being the father and $\frac{q^2}{q^2+2pq}$ chance of being homozygous. If the subject genotype is CC, the father has 100% chance of contributing the C allele and again has $\frac{q^2}{q^2+2pq}$ chance of being homozygous. We therefore expect the relationship to be $\hat{\beta}_C = \frac{1}{2}\hat{\beta}_{CC}\frac{q^2}{q^2+2pq}$, where $\hat{\beta}_C$ is the observed effect per subject allele on father lifespan and $\hat{\beta}_{CC}$ is the reported effect of the homozygous deletion in the father. As before, doubling the allele effect gives an estimate of the effect of the allele on subject lifespan, which finally yielded a converted estimate of 0.155. That is, under Ben-Avrahim *et al.*'s assumptions on inheritance patterns, if their estimate of effect size in minor homozygotes is correct, we should see under the additive model an effect size of 0.155 years, or a $\log_e$(hazard ratio) of –0.015, and correspondingly scaled standard errors (note we are assuming that the effect is actually recessive, and estimating how that effect should appear if an additive model were fitted).

Standard errors were calculated from inferred betas and reported P values, assuming a two-sided test with a normally-distributed estimator. Confidence interval overlap was then assessed using a two-sided test on the estimate difference ($P_{diff}$), using a Z statistic from the difference divided by the standard error of the difference.

## iGWAS

We performed a Bayesian Genome-Wide Association Study using the CES GWAS results and summary statistics on 58 risk factor GWASs (imputed, leading to 7.2 million SNPs in common between all the studies), as described by *McDaid et al. (2017)*. To calculate our prior for SNPs on a given chromosome, first we used a multivariate Mendelian Randomization (masking the focal chromosome) to identify the risk factors significantly influencing lifespan and estimate their causal effect. This identified 16 risk factors independently causally contributing to lifespan (see *Table 2—source data 1* for the causal effect estimates). Next, assuming that a SNP affects lifespan through its effects on the 16 risk factors, prior effects estimates were estimated as the sum of the products of the causal effect estimates of the 16 significant risk factors on lifespan and the effect of the SNP on each risk factor. We added one to the prior effect variance formula described in *McDaid et al. (2017)* to account for the fact that prior effects are estimated using observed Z-scores, and not true Z-scores, with $Z_{obs} \sim \mathcal{N}(Z_{true}, \ 1)$.

We computed Bayes factors by combining the prior effects and the observed association Z statistics. The significance of the Bayes factors was assessed using a permutation approach to calculate P values, by comparing observed Bayes factors to 7.2 billion null Bayes factors. These null Bayes factors were estimated using 1000 null sets of Z statistics combined with the same priors. These empirical P values were then adjusted for multiple testing using the Benjamini-Hochberg procedure.

## Replication in extreme long-livedness

To replicate our novel lifespan findings, we inverse-variance meta-analysed summary statistics from *Deelen et al. (2014)* and *Broer et al. (2015)*, having converted their effect sizes to a common scale (see Candidate SNP replication). These effect sizes were given or could be estimated from P value, effect direction and N, as well as the SNPs MAF.

Both longevity studies and the LifeGen consortium contain individuals from the Rotterdam Study, but due to differences in trait definitions, we could not inflate standard errors directly based on sample overlap. Instead, we calculated the covariance in null SNPs (Z < 1) between each study (r ~ 0.01) and then adjusted the standard errors based on equation 5 from *Lin and Sullivan (2009)*:

$$\text{Variance}(\hat{\beta}) = \text{Variance}_0(\hat{\beta}) + 2 * \sum_{n=1}^{N-1} \sum_{m=n+1}^{N} w_n w_m \, \text{Cov}(\hat{\beta}_n, \hat{\beta}_m)$$

Where $\text{Variance}_0$ is the unadjusted variance of the SNP effect $\hat{\beta}$ after meta-analysis, w is the inverse variance weight of the SNP, N is the number of studies, and Cov $(\hat{\beta}_n, \hat{\beta}_m)$ is the null SNP covariance between each study.

Test of the hypothesis that the effect was zero, was one sided, with alternate hypothesis that the effect had the same sign as in discovery. Effect sizes in discovery and replication were then compared by calculating the ratio (alpha) of replication effect sizes to discovery effect sizes:

$$\alpha = \frac{\beta_{rep}}{\beta_{disc}}$$

and its standard error using the following formula, reflecting the Taylor series expansion of the denominator for SE:

$$\text{SE}_\alpha = \sqrt{\frac{\text{SE}_{rep}^2}{\beta_{disc}^2} + \frac{\beta_{rep}^2 \, \text{SE}_{disc}^2}{\left(\beta_{disc}^2\right)^2}}$$

where *rep* and *disc* are replication and discovery, respectively. Alpha was then inverse-variance meta-analysed across all SNPs to test for collective evidence that the discovery SNPs influence longevity.

## Age and sex-stratified effects

Calculation of age and sex stratified effect sizes was done using the full Cox model (*Equation 1*), imputed dosages and the package 'Survival' in R. We split the full analysis into age decades from 40 to 90 and a wider band, 90–120, beyond that, excluding any parent who died at an age younger than the age band and treating any parent who died beyond the age band as alive at the end of the age band. We thus had, across independent periods of life, estimates of the hazard ratio by decade of age and parent sex, along with standard errors. This gave estimates of the hazard ratio beta(age band, sex) in each age band and sex.

We tested the effect of age and sex, by fitting the linear model beta(age band, sex) = intercept + beta1 x ageband + beta2 x sex + e, where e is independent, but with known variance (the square of the SE in the age/sex stratified model fit) and using the rma function from the R package 'metafor' which uses known variances of dependent variables. The process is more easily understood by examining the age and sex related effect size graphs, and recognising we are fitting age and sex as explanatory variables, considering the standard error of each point shown.

## Causal gene prediction

In order to more accurately implicate causal genes and methylation sites from the detected loci associated with human lifespan, Summary-level Mendelian Randomisation (SMR) and HEterogeneity In InDependent Instruments (HEIDI) tests (*Zhu et al., 2016*) were performed on our CES GWAS statistics. Three separate analyses were performed. First, cis-eQTL scan results from peripheral blood tissue from two previous studies, the Westra data (*Westra et al., 2013*) and CAGE data (*Lloyd-Jones et al., 2017*), were used to prioritize causal genes. Second, cis-eQTL signals (SNPs with FDR < 0.05) for 48 tissues from the GTEx consortium (*Battle et al., 2017*) were used to prioritize causal genes in multiple tissues. Third, genome-wide methylation QTL (mQTL) scan signals in blood tissue from the Brisbane Systems Genetics Study and Lothian Birth Cohort (*McRae et al., 2017*) were used to predict causal CpG sites associated with human lifespan loci. All results from SMR test passing a 5% FDR threshold where the HEIDI test p > 0.05 were reported.

## Fine-mapping using LASSO regression

Selection Operator for JOint multi-SNP analysis (SOJO) (*Ning et al., 2017*) was used to perform conditional fine-mapping analysis of the lifespan loci. The SOJO procedure implements LASSO regression for each locus, which outperforms standard stepwise selection procedure (e.g. GCTA-COJO), based on summary association statistics and the European-ancestry 1000 Genomes samples for LD reference. We based the SOJO analysis on our CES summary association statistics from UK Biobank and used the LifeGen summary statistics as validation sample to optimise the LASSO tuning parameters for each locus. Loci were defined prior to analysis as 1 Mb windows centred at each top variant from the GWAS. For each locus, based on UK Biobank data, we recorded the first 30 variants entering the model and the tuning parameters for these entering points along the LASSO path, as well as the LASSO results at the tuning parameters. For each recorded tuning parameter, we then built a polygenic score and computed the proportion of predictable variance from the regional polygenic score in the validation sample. The best out-of-sample R squared is reported, together with the selected variants per locus.

## Lifespan variance explained

We sought an independent set of disease-associated SNPs to assess which diseases had the greatest effect on lifespan. A large number of SNPs per disease category, especially other cancers, were used to ensure that diseases were not under-represented when testing for association with lifespan. The latest, genome-wide significant disease SNPs from European ancestry studies were retrieved from the GWAS catalog (14 March 2018), based on string matching within reported trait names. For Alzheimer's/Parkinson's disease, these were 'alzh' and 'parkin'; for CVD, these were 'myocard', 'cvd', 'cardiovascular', 'coronary', and 'artery disease'; for Type 2 diabetes this was 'type 2 diabetes'; for cancers, this was 'cancer', 'noma', 'ioma', 'tumo[u]r', and 'leukemia'. Cancers were then divided in Lung cancer and Other cancers based on the presence or absence of the keyword 'lung'. The Smoking/Lung cancer category was created by adding traits containing the keywords 'smoking' and 'chronic obstructive' to the lung cancers. Each category was manually checked to include only

associations with the diseases themselves or biomarkers of the diseases. Although some throat cancers are often caused by smoking and alcohol consumption, we did not treat these as smoking loci; in practice, this choice had no effect as the only significant throat cancer locus (oesophageal cancer near CFTR) was discounted as secondary pleiotropic – see below.

SNPs missing from the CES summary statistics were imputed from the closest proxy (min. $r^2 > 0.9$) or averaged from multiple proxies if equally close. SNPs without effect sizes, SNPs matching neither our reference nor effect alleles, and SNPs with reported frequencies differing by more than 0.3 from our own were excluded. The remaining SNPs were subdivided into independent ($r^2 < 0.1$) loci 500 kb apart, keeping the SNP most strongly associated with lifespan in the CES GWAS – thus proportional only to the lifespan GWAS test statistic and independent from the number of disease SNPs per category. Lastly, where possible, loci were tested for association with their disease category in UK Biobank, using self-reported diseases of 325,292 unrelated, genomically British subjects, their siblings, and each parent separately. Diseases of subject relatives were already coded into broad disease categories by UK Biobank. For subjects themselves, ICD codes had been recorded which we grouped into similar categories (hypertension, cerebral infarction, heart disease, diabetes, dementia, depression, stress, pulmonary disease, and cancer, in accordance with *Figure 10—source data 1*, although cancer in subjects was more directly taken as the trait of reporting number of cancers > 0). The trait of reporting these diseases (separately for each relative and the subject themselves) was then tested for association with genotypic dosage for the GWAS catalog disease SNPs. The model fitted was a logistic regression of not reporting the disease, using the same covariates as the CES GWAS with the addition of subject age, and estimated the log odds ratio of protection from disease in UK Biobank for each copy of the disease-protective allele in the GWAS catalog. Effects reported in the GWAS catalog for which we found the pooled estimate from our association study was in the opposite direction were flipped (if p < 0.05) or discarded (if p ≥ 0.05).

Our final dataset consisted of 555 disease SNPs (81 neurological, 72 cardiovascular, 65 diabetes, 22 smoking/lung cancer, and 315 other cancers). Lifespan variance explained (LVE) was calculated as $2pqa^2$, where p and q are the frequencies of the effect and reference alleles in our lifespan GWAS, and a is the SNP effect size in years of life (*Falconer et al., 1996*). To assess pleiotropy, SNPs were tested against other disease categories, and where possible, the relative strengths of standardised associations between disease categories were compared. SNPs associating more strongly with another disease, as defined by a Z statistic more than double that of the original disease, were marked as pleiotropic and secondary. Whilst strength of association would not normally be perceived as appropriately measured in this way (odds ratio being more conventional and independent of prevalence), here we are interested in the excess number of disease cases in the population due to the variant, so any locus with a moderate OR for a highly prevalent disease is judged more causative of that disease than a locus with a (somewhat) higher OR for a very rare disease, as the number of attributable cases will be lower. The Z statistic captures this – given that p and q are obviously the same (same SNP, same population). Correspondingly, for diseases only present in one sex, the other sex was treated as all controls. Whilst this halves the apparent effect size, the required measure is the amount of disease caused across the whole population. A SNP conferring similar attributable counts of CVD and breast cancer in women, but also CVD in men, is causing CVD more than cancer across the population. Correspondingly, selection pressure on the breast cancer effect is half that for a matching effect in both sexes. SNPs conferring both an increase in disease and an increase in lifespan were marked as antagonistically pleiotropic. Unsurprisingly, in practice, there were one or more other diseases reduced by the SNP and therefore the reported disease-increasing association was considered secondary. Total LVE per disease category was calculated by summing SNPs not marked as secondary and with significant effects on lifespan, where significance was determined by setting an FDR threshold that allowed for one false positive among all SNPs tested (q ≤ 0.016, 60 SNPs). To compare the cumulative LVE of the top LVE loci, all non-secondary association SNPs from the disease categories were pooled and again subdivided into independent loci ($r^2 < 0.1$) 500 kb apart. Applying an FDR threshold with the same criteria (q ≤ 0.022), a total of 45 (1 neurological, 23 cardiovascular, four diabetes, six smoking/lung cancer, and 11 other cancer) independent loci remained and their LVE was summed by disease category.

## Cell type and pathway enrichment

Stratified LD-score regression (*Finucane et al., 2015*) partitions SNP heritability into regions linked to specific tissues and cell types, such as super-enhancers and histone marks, and then assesses whether the SNPs in these regions contribute disproportionately to the total SNP heritability. Standard LD-score regression (*Bulik-Sullivan et al., 2015*) indicated that between the different samples (UK Biobank and/or LifeGen) and analyses (CES or SSE), the CES results from UK Biobank genomically British ancestry individuals had the highest SNP heritability, plausibly due to its uniformity of population sample. These statistics were analysed using the procedure described by *Finucane et al., 2015*, which included limiting the regressions to HapMap3 SNPs with MAF > 0.05 to reduce statistical noise. Results from all cell types were merged and then adjusted for multiple testing using Benjamini–Hochberg (FDR 5%).

The full CES dataset was subjected to gene-based tests, which used up to $10^6$ SNP permutations per gene to assign P values to 26,056 genes, as implemented by VEGAS2 v2.01.17 (*Mishra and Macgregor, 2015*). Only directly genotyped SNPs from the UK Biobank array were used to facilitate practical runtimes. Using the default settings, all SNPs located within genes (relative to the 5' and 3' UTR) were included. Scored genes were then tested for enrichment in 9741 pathways from the NCBI BioSystems Database with up to $10^8$ gene permutations per pathway using VEGAS2Pathway (*Mishra and MacGregor, 2017*). Pathway enrichment P values were automatically adjusted for pathway size (empirical P) and further adjusted for multiple testing using Benjamini-Hochberg (FDR 5%).

DEPICT was also used to create a list of genes; however, this method uses independent SNPs passing a P value threshold to define lifespan loci and then attempts to map 18,922 genes to them. Gene prioritization and subsequent gene set enrichment is done for 14,461 probabilistically-defined reconstituted gene sets, which are tested for enrichment under the self-contained null hypothesis (*Pers et al., 2015*). Two separate analyses were performed on the CES summary statistics, using independent SNPs (>500 kb between top SNPs) which were present in the DEPICT database. The first analysis used a genome-wide significance threshold (GW DEPICT analysis) and mapped genes to 10 loci, automatically excluding the major histocompatibility complex (MHC) region. The second used a suggestive significance threshold ($p < 10^{-5}$), which yielded 93 loci and mapped genes to 91 of these, again excluding the MHC region. To test if pathways were significantly enriched at a 5% FDR threshold, we used the values calculated by DEPICT, already adjusted for the non-independence of the gene sets tested.

PASCAL was used with the same summary statistics and gene sets as DEPICT, except the gene probabilities within the sets were dichotomized (Z > 3) (*Marouli et al., 2017*), leading to the analysis of the same 14,461 pathways. PASCAL transformed SNP P values into gene-based P values (with default method '–genescoring=sum') for 21,516 genes (*Lamparter et al., 2016*). When testing the pathways for overrepresentation of high gene scores, the P values are estimated under the competitive null hypothesis (*Maciejewski, 2014*). These pathway empirical P values were further adjusted for multiple testing using Benjamini-Hochberg procedure.

## Age-related eQTLs enrichment

We identified SNPs in our CES GWAS that were eQTLs that is associated with the expression of at least one gene with $p < 10^{-4}$ in a dataset by the eQTLGen Consortium (n = 31,684 individuals) (*Võsa et al., 2018*). A total of 3577 eQTLs after distance pruning (500 kb) were present, of which 755 were associated with genes differentially expressed with age (*Peters et al., 2015*). We used Fisher's exact test to determine, amongst the set of eQTLs, if SNPs which were associated with lifespan (at varying thresholds of statistical significance) were enriched for SNPs associated with genes whose expression is age-related.

## Polygenic lifespan score associations

We used the CES GWAS, excluding (one at a time) all Scottish populations (whether from Scottish UK Biobank assessment centres or Scottish LifeGen cohorts), Estonian populations and a random 10% of UK Biobank English and Welsh subjects to create polygenic risk scores using PRSice (*Euesden et al., 2015*), where the test subjects had not been part of the training data. As we find polygenic risk scores developed using all ($p \leq 1$) independent ($r^2 < 0.1$) SNPs (PRSP1), rather than

those passing a tighter significance threshold are most associated (highest standardised effect size; see *Table 3—source data 1* for comparison between thresholds), these were used in the analysis.

To make cross-validated lifespan associations using polygenic scores, our unrelated, genomically British sample was partitioned into training and test sets. The first test set consisted of Scottish individuals from UK Biobank, as defined by assessment centre or northings and eastings falling within Scotland (N = 24,059). The second set consisted of a random subset of the remaining English and Welsh population, reproducibly sampled based on the last digit of their UK Biobank identification number (#7, N = 29,815). The training set was constructed by excluding these two populations, as well as excluding individuals from Generation Scotland, from our GWAS and recalculating estimates of beta on that subset.

A third independent validation set was constructed by excluding the EGCUT cohort from the Life-Gen sample and using the remaining data to test lifespan in the newly genotyped EGCUT cohort (*Leitsalu et al., 2015*), using unrelated individuals only (N = 36,499).

Polygenic survival scores were constructed using PRSice 2.0.14.beta (*Euesden et al., 2015*) in a two-step process. First, lifespan SNPs were LD-clumped based on an $r^2$ threshold of 0.1 and a window size of 250 kb. To facilitate practical run times of PRSice clumping, only directly genotyped SNPs were used in the Scottish and English/Welsh subsets. The Estonian sample was genotyped on four different arrays with limited overlap, so here imputed data (with imputation measure R2 > 0.9) was used and clumped with PLINK directly ($r^2$ = 0.1; window = 1000 kb). The clumped SNPs (85,539 in UK Biobank, 68,234 in Estonia) were then further pruned based on several different P value thresholds, to find the most informative subset. For all individuals, a polygenic score was calculated as the sum of SNP dosages (of SNPs passing the P value threshold) multiplied by their estimated allele effect. These scores were then standardised to allow for associations to be expressed in standard deviations in polygenic scores.

Polygenic scores of test cohorts were regressed against lifespan and alive/dead status using a cox proportional hazards model, adjusted for sex, assessment centre, batch, array, and 10 principal components. Where parental lifespan was used, hazard ratios were doubled to gain an estimate of the polygenic score on own mortality. Scores were also regressed against self-reported diseases in UK Biobank subjects, their siblings, and each parent separately, using a logistic regression adjusted for the same covariates as in the lifespan analysis plus subject age. As with previous disease associations, estimates were transformed so positive associations indicate a protective or life-extending effect, and effect estimates of first degree relatives were doubled. Meta-analysis of estimates between cohorts was done using inverse variance weighting. Where estimates between kin were meta-analysed, standard errors were adjusted for correlation between family members. This involved multiplying standard errors by $\sqrt{1+r}$ for each correlation (r) with the reference kin (*Equation 2*), which appears slightly conservative. As correlations between family member diseases were very low (range 0.0005 to 0.1048), in practice, this adjustment had no effect.

## URLs

MultiABEL: https://github.com/xiashen/MultiABEL/
LDSC: https://github.com/bulik/ldsc
SMR/HEIDI: https://cnsgenomics.com/software/smr/
SOJO: https://github.com/zhenin/sojo/
DEPICT: https://www.broadinstitute.org/mpg/depict/
PASCAL: https://www2.unil.ch/cbg/index.php?title=Pascal
GTEx: https://gtexportal.org/home/datasets

## Acknowledgments

We thank the UK Biobank Resource, approved under application 8304; we acknowledge funding from the UK Medical Research Council Human Genetics Unit, Wellcome Trust PhD Training Fellowship for Clinicians - the Edinburgh Clinical Academic Track (ECAT) programme (204979/Z/16/Z), the Medical Research Council Doctoral Training Programme in Precision Medicine (MR/N013166/1) and the AXA research fund. We thank Tom Haller of the University of Tartu, for tailoring RegScan so we could use it with compressed files (*Haller et al., 2015*). We would also like to thank the researchers, funders and participants of the LifeGen consortium (*Joshi et al., 2017*).

# Additional information

## Group author details

**eQTLGen Consortium**

**M Agbessi**: Ontario Institute for Cancer Research, Toronto, Canada; **H Ahsan**: Department of Public Health Sciences, University of Chicago, Chicago, United States; **I Alves**: Ontario Institute for Cancer Research, Toronto, Canada; **A Andiappan**: Singapore Immunology Network, Agency for Science, Technology and Research, Singapore, Singapore; **P Awadalla**: Ontario Institute for Cancer Research, Toronto, Canada; **A Battle**: Department of Computer Science, Johns Hopkins University, Baltimore, United States; **MJ Bonder**: Department of Genetics University, Medical Centre Groningen, Groningen, The Netherlands; **D Boomsma**: Vrije Universiteit, Amsterdam, The Netherlands; **M Christiansen**: Cardiovascular Health Research Unit, University of Washington, Seattle, United States; **A Claringbould**: Department of Genetics University, Medical Centre Groningen, Groningen, The Netherlands; **P Deelen**: Department of Genetics University, Medical Centre Groningen, Groningen, The Netherlands; **J van Dongen**: Vrije Universiteit, Amsterdam, The Netherlands; **T Esko**: Estonian Genome Center, University of Tartu, Tartu, Estonia; **M Favé**: Ontario Institute for Cancer Research, Toronto, Canada; **L Franke**: Department of Genetics University, Medical Centre Groningen, Groningen, The Netherlands; **T Frayling**: Exeter Medical School, University of Exeter, Exeter, United Kingdom; **SA Gharib**: Department of Medicine, University of Washington, Seattle, United States; **G Gibson**: School of Biological Sciences, Georgia Institute of Technology, Atlanta, United States; **G Hemani**: MRC Integrative Epidemiology Unit, University of Bristol, Bristol, United Kingdom; **R Jansen**: Vrije Universiteit, Amsterdam, The Netherlands; **A Kalnapenkis**: Estonian Genome Center, University of Tartu, Tartu, Estonia; **S Kasela**: Estonian Genome Center, University of Tartu, Tartu, Estonia; **J Kettunen**: University of Helsinki, Helsinki, Finland; **Y Kim**: Department of Computer Science, Johns Hopkins University, Baltimore, United States; **H Kirsten**: Institut für Medizinische Informatik, Statistik und Epidemiologie, LIFE – Leipzig ResearchCenter for Civilization Diseases, Universität Leipzig, Leipzig, Germany; **P Kovacs**: IFB Adiposity Diseases, Universität Leipzig, Leipzig, Germany; **K Krohn**: Interdisciplinary Center for Clinical Research, Faculty of Medicine, Universität Leipzig, Leipzig, Germany; **J Kronberg-Guzman**: Estonian Genome Center, University of Tartu, Tartu, Estonia; **V Kukushkina**: Estonian Genome Center, University of Tartu, Tartu, Estonia; **Z Kutalik**: Lausanne University Hospital, Lausanne, Switzerland; **M Kähönen**: Department of Clinical Physiology and Faculty of Medicine and Life Sciences, Tampere University Hospital and University of Tampere, Tampere, Finland; **B Lee**: Singapore Immunology Network, Agency for Science, Technology and Research, Singapore, Singapore; **T Lehtimäki**: Department of Clinical Chemistry, Fimlab Laboratories and Faculty of Medicine and Life Sciences, University of Tampere, Tampere, Finland; **M Loeffler**: Institut für Medizinische Informatik, Statistik und Epidemiologie, LIFE – Leipzig Research Center for Civilization Diseases, Universität Leipzig, Leipzig, Germany; **U Marigorta**: School of Biological Sciences, Georgia Institute of Technology, Atlanta, United States; **A Metspalu**: Estonian Genome Center, University of Tartu, Tartu, Estonia; **J van Meurs**: Department of Internal Medicine, Erasmus Medical Centre, Rotterdam, The Netherlands; **L Milani**: Estonian Genome Center, University of Tartu, Tartu, Estonia; **M Müller-Nurasyid**: Institute of Genetic Epidemiology, Helmholtz Zentrum München, German Research Center for Environmental Health, Neuherberg, Germany; **M Nauck**: Institute of Clinical Chemistry and Laboratory Medicine, University Medicine Greifswald, Greifswald, Germany; **M Nivard**: Vrije Universiteit, Amsterdam, The Netherlands; **B Penninx**: Vrije Universiteit, Amsterdam, The Netherlands; **M Perola**: National Institute for Health and Welfare, University of Helsinki, Helsinki, Finland; **N Pervjakova**: Estonian Genome Center, University of Tartu, Tartu, Estonia; **B Pierce**: Department of Public Health Sciences, University of Chicago, Chicago, United States; **J Powell**: Institute for Molecular Bioscience, University of Queensland, Brisbane, Australia; **H Prokisch**: Institute of Human Genetics, Helmholtz Zentrum München, München, Germany; **BM Psaty**: Departments of Epidemiology, Medicine, and Health Services, Cardiovascular Health Research Unit, University of Washington, Seattle, United States; **O Raitakari**: Department of Clinical Physiology and Nuclear Medicine, Turku University Hospital and University of Turku, Turku, Finland; **S Ring**: School of Social and Community Medicine, University of Bristol, Bristol, United Kingdom; **S Ripatti**: University of Helsinki, Helsinki, Finland; **O Rotzschke**: Singapore Immunology Network, Agency for Science, Technology and Research, Singapore, Singapore; **S**

Ruëger: Lausanne University Hospital, Lausanne, Switzerland; A Saha: Department of Computer Science, Johns Hopkins University, Baltimore, United States; M Scholz: Institut für Medizinische InformatiK, Statistik und Epidemiologie, LIFE – Leipzig Research Center for Civilization Diseases, Universität Leipzig, Leipzig, Germany; K Schramm: Institute of Genetic Epidemiology, Helmholtz Zentrum München, German Research Center for Environmental Health, Neuherberg, Germany; I Seppälä: Department of Clinical Chemistry, Fimlab Laboratories and Faculty of Medicine and Life Sciences, University of Tampere, Tampere, Finland; M Stumvoll: Department of Medicine, Universität Leipzig, Leipzig, Germany; P Sullivan: Department of Medical Epidemiology and Biostatistics, Karolinska Institutet, Stockholm, Sweden; A Teumer: Institute for Community Medicine, University Medicine Greifswald, Greifswald, Germany; J Thiery: Institute for Laboratory Medicine, LIFE – Leipzig Research Center for Civilization Diseases, Universität Leipzig, Leipzig, Germany; L Tong: Department of Public Health Sciences, University of Chicago, Chicago, United States; A Tönjes: Department of Medicine, Universität Leipzig, Leipzig, Germany; J Verlouw: Department of Internal Medicine, Erasmus Medical Centre, Rotterdam, The Netherlands; PM Visscher: Institute for Molecular Bioscience, University of Queensland, Brisbane, Australia; U Võsa: Department of Genetics, University Medical Centre Groningen, Groningen, The Netherlands; U Völker: Interfaculty Institute for Genetics and Functional Genomics, University Medicine Greifswald, Greifswald, Germany; H Yaghootkar: Exeter Medical School, University of Exeter, Exeter, United Kingdom; J Yang: Institute for Molecular Bioscience, University of Queensland, Brisbane, Australia; B Zeng: School of Biological Sciences, Georgia Institute of Technology, Atlanta, United States; F Zhang: Institute for Molecular Bioscience, University of Queensland, Brisbane, Australia; M Agbessi: Ontario Institute for Cancer Research, Toronto, Canada; H Ahsan: Department of Public Health Sciences, University of Chicago, Chicago, United States; I Alves: Ontario Institute for Cancer Research, Toronto, Canada; A Andiappan: Singapore Immunology Network, Agency for Science, Technology and Research, Singapore, Singapore; P Awadalla: Ontario Institute for Cancer Research, Toronto, Canada; A Battle: Department of Computer Science, Johns Hopkins University, Baltimore, United States; MJ Bonder: Department of Genetics University, Medical Centre Groningen, Groningen, The Netherlands; D Boomsma: Vrije Universiteit, Amsterdam, The Netherlands; M Christiansen: Cardiovascular Health Research Unit, University of Washington, Seattle, United States; A Claringbould: Department of Genetics University, Medical Centre Groningen, Groningen, The Netherlands; P Deelen: Department of Genetics University, Medical Centre Groningen, Groningen, The Netherlands; J van Dongen: Vrije Universiteit, Amsterdam, The Netherlands; T Esko: Estonian Genome Center, University of Tartu, Tartu, Estonia; M Favé: Ontario Institute for Cancer Research, Toronto, Canada; L Franke: Department of Genetics University, Medical Centre Groningen, Groningen, The Netherlands; T Frayling: Exeter Medical School, University of Exeter, Exeter, United Kingdom; SA Gharib: Department of Medicine, University of Washington, Seattle, United States; G Gibson: School of Biological Sciences, Georgia Institute of Technology, Atlanta, United States; G Hemani: MRC Integrative Epidemiology Unit, University of Bristol, Bristol, United Kingdom; R Jansen: Vrije Universiteit, Amsterdam, The Netherlands; A Kalnapenkis: Estonian Genome Center, University of Tartu, Tartu, Estonia; S Kasela: Estonian Genome Center, University of Tartu, Tartu, Estonia; J Kettunen: University of Helsinki, Helsinki, Finland; Y Kim: Department of Computer Science, Johns Hopkins University, Baltimore, United States; H Kirsten: Institut für Medizinische Informatik, Statistik und Epidemiologie, LIFE – Leipzig ResearchCenter for Civilization Diseases, Universität Leipzig, Leipzig, Germany; P Kovacs: IFB Adiposity Diseases, Universität Leipzig, Leipzig, Germany; K Krohn: Interdisciplinary Center for Clinical Research, Faculty of Medicine, Universität Leipzig, Leipzig, Germany; J Kronberg-Guzman: Estonian Genome Center, University of Tartu, Tartu, Estonia; V Kukushkina: Estonian Genome Center, University of Tartu, Tartu, Estonia; Z Kutalik: Lausanne University Hospital, Lausanne, Switzerland; M Kähönen: Department of Clinical Physiology and Faculty of Medicine and Life Sciences, Tampere University Hospital and University of Tampere, Tampere, Finland; B Lee: Singapore Immunology Network, Agency for Science, Technology and Research, Singapore, Singapore; T Lehtimäki: Department of Clinical Chemistry, Fimlab Laboratories and Faculty of Medicine and Life Sciences, University of Tampere, Tampere, Finland; M Loeffler: Institut für Medizinische Informatik, Statistik und Epidemiologie, LIFE – Leipzig Research Center for Civilization Diseases, Universität Leipzig, Leipzig, Germany; U Marigorta: School of Biological Sciences, Georgia Institute of Technology, Atlanta, United States; A Metspalu: Estonian Genome Center, University of Tartu, Tartu, Estonia; J van Meurs: Department of Internal

Medicine, Erasmus Medical Centre, Rotterdam, The Netherlands; **L Milani**: Estonian Genome Center, University of Tartu, Tartu, Estonia; **M Müller-Nurasyid**: Institute of Genetic Epidemiology, Helmholtz Zentrum München, German Research Center for Environmental Health, Neuherberg, Germany; **M Nauck**: Institute of Clinical Chemistry and Laboratory Medicine, University Medicine Greifswald, Greifswald, Germany; **M Nivard**: Vrije Universiteit, Amsterdam, The Netherlands; **B Penninx**: Vrije Universiteit, Amsterdam, The Netherlands; **M Perola**: National Institute for Health and Welfare, University of Helsinki, Helsinki, Finland; **N Pervjakova**: Estonian Genome Center, University of Tartu, Tartu, Estonia; **B Pierce**: Department of Public Health Sciences, University of Chicago, Chicago, United States; **J Powell**: Institute for Molecular Bioscience, University of Queensland, Brisbane, Australia; **H Prokisch**: Institute of Human Genetics, Helmholtz Zentrum München, München, Germany; **BM Psaty**: Departments of Epidemiology, Medicine, and Health Services, Cardiovascular Health Research Unit, University of Washington, Seattle, United States; **O Raitakari**: Department of Clinical Physiology and Nuclear Medicine, Turku University Hospital and University of Turku, Turku, Finland; **S Ring**: School of Social and Community Medicine, University of Bristol, Bristol, United Kingdom; **S Ripatti**: University of Helsinki, Helsinki, Finland; **O Rotzschke**: Singapore Immunology Network, Agency for Science, Technology and Research, Singapore, Singapore; **S Ruёger**: Lausanne University Hospital, Lausanne, Switzerland; **A Saha**: Department of Computer Science, Johns Hopkins University, Baltimore, United States; **M Scholz**: Institut für Medizinische InformatiK, Statistik und Epidemiologie, LIFE – Leipzig Research Center for Civilization Diseases, Universität Leipzig, Leipzig, Germany; **K Schramm**: Institute of Genetic Epidemiology, Helmholtz Zentrum München, German Research Center for Environmental Health, Neuherberg, Germany; **I Seppälä**: Department of Clinical Chemistry, Fimlab Laboratories and Faculty of Medicine and Life Sciences, University of Tampere, Tampere, Finland; **M Stumvoll**: Department of Medicine, Universität Leipzig, Leipzig, Germany; **P Sullivan**: Department of Medical Epidemiology and Biostatistics, Karolinska Institutet, Stockholm, Sweden; **A Teumer**: Institute for Community Medicine, University Medicine Greifswald, Greifswald, Germany; **J Thiery**: Institute for Laboratory Medicine, LIFE – Leipzig Research Center for Civilization Diseases, Universität Leipzig, Leipzig, Germany; **L Tong**: Department of Public Health Sciences, University of Chicago, Chicago, United States; **A Tönjes**: Department of Medicine, Universität Leipzig, Leipzig, Germany; **J Verlouw**: Department of Internal Medicine, Erasmus Medical Centre, Rotterdam, The Netherlands; **PM Visscher**: Institute for Molecular Bioscience, University of Queensland, Brisbane, Australia; **U Võsa**: Department of Genetics, University Medical Centre Groningen, Groningen, The Netherlands; **U Völker**: Interfaculty Institute for Genetics and Functional Genomics, University Medicine Greifswald, Greifswald, Germany; **H Yaghootkar**: Exeter Medical School, University of Exeter, Exeter, United Kingdom; **J Yang**: Institute for Molecular Bioscience, University of Queensland, Brisbane, Australia; **B Zeng**: School of Biological Sciences, Georgia Institute of Technology, Atlanta, United States; **F Zhang**: Institute for Molecular Bioscience, University of Queensland, Brisbane, Australia

## Competing interests

Tõnu Esko: Reviewing Editor, eLife. The other authors declare that no competing interests exist.

## Funding

| Funder | Grant reference number | Author |
|---|---|---|
| Medical Research Council | DTP in Precision Medicine MR/N013166/1,HGU QTL in health and disease | Paul RHJ Timmers Andrew D Bretherick David W ClarkeQTLGen ConsortiumJames F Wilson |
| Estonian Research Competency Council | PUT 1665 | Kristi Lall Krista Fischer |
| Wellcome Trust | PhD Training Fellowship for Clinicians | Andrew D Bretherick |
| Edinburgh Clinical Academic Track | 204979/Z/16/Z | Andrew D Bretherick |
| Svenska Forskningsrådet Formas | 2014-00371 | Xia Shen |

| Svenska Forskningsrådet Formas | 2017-02543 | Xia Shen |
| Schweizerischer Nationalfonds zur Förderung der Wissenschaftlichen Forschung | 31003A_169929 | Zoltán Kutalik |
| SystemsX.ch | 51RTP0_151019 | Zoltán Kutalik |
| AXA Research Fund | | Peter K Joshi |

The funders had no role in study design, data collection and interpretation, or the decision to submit the work for publication.

### Author contributions

Paul RHJ Timmers, Ninon Mounier, Formal analysis, Investigation, Visualization, Writing—original draft; Kristi Lall, Formal analysis, Investigation, Writing—original draft; Krista Fischer, Xia Shen, Conceptualization, Formal analysis, Supervision, Investigation, Visualization, Methodology, Writing—original draft; Zheng Ning, Xiao Feng, Formal analysis, Visualization, Writing—review and editing; Andrew D Bretherick, David W Clark, Software, Formal analysis, Writing—review and editing; Tõnu Esko, Resources, Supervision, Funding acquisition, Writing—original draft; Zoltán Kutalik, Conceptualization, Resources, Software, Investigation, Methodology, Writing—original draft; James F Wilson, Conceptualization, Resources, Supervision, Funding acquisition, Writing—original draft; Peter K Joshi, Conceptualization, Supervision, Validation, Methodology, Writing—original draft, Writing—review and editing

### Author ORCIDs

Paul RHJ Timmers http://orcid.org/0000-0002-5197-1267
Xia Shen http://orcid.org/0000-0003-4390-1979
Tõnu Esko http://orcid.org/0000-0003-1982-6569
James F Wilson http://orcid.org/0000-0001-5751-9178
Peter K Joshi http://orcid.org/0000-0002-6361-5059

### Ethics

Human subjects: This work used existing datasets, for which ethical approval had been gathered for genetic investigation at the time of collection.

### Decision letter and Author response

Decision letter https://doi.org/10.7554/eLife.39856.047
Author response https://doi.org/10.7554/eLife.39856.048

## Additional files

### Supplementary files

• Supplementary file 1. Loci with significantly predicted candidate genes using SMR-HEIDI test and two eQTL datasets (blood tissue). This PDF contains a table and plots of the lifespan GWAS and eQTL signals genes from Westra and CAGE eQTL studies that pass FDR < 5% threshold for the SMR test and p>0.05 threshold for HEIDI test.
DOI: https://doi.org/10.7554/eLife.39856.035

• Supplementary file 2. Predicted causal elements by SMR-HEIDI using expression and methylation QTL data. This Excel workbook contains two sheets. The first sheet lists the genes and tissues prioritised by SMR-HEIDI using GTEx expression data, while the second sheet lists the methylation sites prioritised using mQTL data.
DOI: https://doi.org/10.7554/eLife.39856.036

• Supplementary file 3. Evidence of allelic heterogeneity of the lifespan loci via identification of secondary associations using SOJO. This Excel workbook contains two sheets. The first sheet is a summary of the number of additional variants identified by SOJO per locus. The second sheet is a

detailed list of all variants identified by SOJO per locus, their allele frequencies, and their independent effects.

DOI: https://doi.org/10.7554/eLife.39856.037

• Supplementary file 4. Cell types enriched for lifespan heritability identified by stratified LD-score regression. This Excel workbook contains a list of all cell types tested for lifespan heritability enrichment by Stratified LD-score regression and includes a list of coefficients for each cel type.

DOI: https://doi.org/10.7554/eLife.39856.038

• Supplementary file 5. Gene sets highlighted by VEGAS2Pathway and corresponding results from DEPICT and PASCAL (FDR < 5%). This Excel workbook contains two sheets. The first sheet lists the gene sets highlighted by VEGAS2Pathway as enriched for lifespan genes at FDR 5%, with corresponding results for DEPICT and PASCAL enrichment analyses. The second sheet lists all VEGAS gene sets significant at FDR < 15%, with a list of genes included in each set.

DOI: https://doi.org/10.7554/eLife.39856.039

• Supplementary file 6. eQTL SNPs associated with lifespan for genes whose expression varies with age. This PDF contains a table and figure on the age-related gene expression analysis.

DOI: https://doi.org/10.7554/eLife.39856.040

• Transparent reporting form

DOI: https://doi.org/10.7554/eLife.39856.041

## Data availability

Individual phenotypic and genetic data is available from UK Biobank upon application: https://www.ukbiobank.ac.uk/register-apply/. Phenotypes used in this work include subject age, sex, ethnicity, relatedness, genotyping batch, array, and principal components, as well as parental age and alive/dead status. Also included are self-reported diseases of subjects and their families. A full description of our application can be found at https://www.ukbiobank.ac.uk/2015/02/dr-james-wilson-university-of-edinburgh-centre-for-population-sciences/. The results that support our findings, in particular, the GWAS summary statistics we generated for >1 million parental lifespans in this study are available at http://dx.doi.org/10.7488/ds/2463. Gene expression data used in the age-related gene analysis is being made available by the eQTLGen Consortium, see http://www.eqtlgen.org/ and Võsa, U. et al. Unraveling the polygenic architecture of complex traits using blood eQTL meta-analysis. bioRxiv 447367 (2018). Single tissue gene expression data used in the SMR-Heidi analysis can be found on the GTEx website https://gtexportal.org/home/datasets, under GTEx_Analysis_v7_eQTL.tar.gz. Source data files for main figures and tables are included as supplementary material, other than GWAS summary statistics.

The following dataset was generated:

| Author(s) | Year | Dataset title | Dataset URL | Database and Identifier |
|---|---|---|---|---|
| Timmers PRHJ, Läll K, Fischer K, Ning Z, Feng X, Bretherick A, Clark DW, Shen X, Xia Esko T, Kutalik Z, Wilson JF, Joshi PK | 2018 | Genomic underpinnings of lifespan allow prediction and reveal basis in modern risks | http://dx.doi.org/10.7488/ds/2463 | Edinburgh DataShare, 10.7488/ds/2463 |

The following previously published dataset was used:

| Author(s) | Year | Dataset title | Dataset URL | Database and Identifier |
|---|---|---|---|---|
| GTEx Consortium | 2017 | Single tissue gene expression data used in the SMR-Heidi analysis | https://gtexportal.org/home/datasets | Genotype-Tissue Expression, GTEx_Analysis_v7_eQTL.tar.gz |

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
