## [Decision Letter]

[**Editorial note:** This article has been through an editorial process in which the authors decide how to respond to the issues raised during peer review. The Reviewing Editor's assessment is that all the issues have been addressed.]

Thank you for submitting your article "Genomic underpinnings of lifespan allow prediction and reveal basis in modern risks" for consideration by *eLife*. Your article has been reviewed by three peer reviewers, one of whom is a member of our Board of Reviewing Editors, and the evaluation has been overseen by Mark McCarthy as the Senior Editor. The following individual involved in review of your submission has agreed to reveal his identity: Joris Deelen (Reviewer #3). The other reviewers remain anonymous.

The Reviewing Editor has highlighted the concerns that require revision and/or responses, and we have included the separate reviews below for your consideration. If you have any questions, please do not hesitate to contact us.

The major concerns are as follows:

– We found the paper was often hard to read and would benefit from substantial re-writing to improve clarity, being more succinct at some points, and providing additional necessary details at other occasions.

– We recommend that the authors limit the discussion around the (non)replication of results, and focus more attention on the subset of loci that show significant association.

– We felt that (as has been shown repeatedly) a one-stage approach – using a more stringent significance threshold – would likely provide more statistical power for discovery, than the current multiple-stage design. It would also simplify and focus the reporting, allowing more in-depth analysis and discussion of each (robustly) associated locus.

– We recommend that UKBiobank analyses use appropriate statistical tools to account for population stratification and/or (cryptic) relatedness (e.g. BOLT-LMM); and that any overlap between samples is removed when combining data.

– We recommend that analyses indicating significant sex-specific effects include both sex-interaction and sex-stratified analyses.

– We felt strongly that there was an opportunity for more in-depth discussion to increase the informativeness of the study: see reviewer comments for suggestions.

– An important topic that has not been fully worked out relates to the proportion of the genetic basis of longevity that is explained by avoiding disease vs other mechanisms; i.e. is longevity simply about avoiding disease?

Separate reviews (please include a response on each point):

*Reviewer #1:*

This study reports on the genetic underpinnings of lifespan using survival data from 635,205 parents of participants to the UK Biobank (British ancestry only) and of 377,035 parents of participants to the LifeGen of European ancestry. A total of 6 novel loci were identified, whereas another 6 loci from previous studies were replicated. Nevertheless, several other loci reported before for longevity were not replicated in the current study for lifespan. Consistent with previous reports, the life-span increasing alleles of several of the lead variants has protective effects on cardiovascular disease (risk factors), type 2 diabetes, COPD, and lung cancer. Surprisingly, lifespan increasing alleles were not associated with lower risk of others cancers. Pathway enrichment analyses confirmed the importance of pathways related to lipoprotein particles, but also highlighted new pathways involved in vesicle transport, metabolism of acylglycerol and sterols and synaptic and dendritic function. Using a polygenic risk score, based on the lifespan loci, the authors estimate that individuals in the top decile, carriers the most lifespan increasing alleles, live on average 5 years longer than individuals in the bottom decile of individuals (with the fewest lifespan increasing alleles).

General comments

Strengths:

The authors have combined a massive amount of information, using a kin-cohort design, genotype information of participants and phenotype data of parents are used for analyses. In the context of longevity, this approach allows using more complete information as compared to using phenotype (age of death) data of participants themselves.

Weaknesses:

The paper is generally hard to read; the text is often long-winded, yet, at many occasions, critical details are missing for readers to be able to interpret results. Along the same lines, some interpretations by the authors lack background for readers to be able to follow their reasoning. The paper will require a substantial re-writing to get the valuable information across. I have provided some suggestions in the "specific comments" section.

The study design is confusing and seems generally inefficient.

– For example, why were >70,000 related individuals (i.e. >17% of the population) removed from analyses – methods exist to use the information of related individuals (e.g. BOLT-LMM).

– The kin-cohort is an interesting design; how does deal with (or take advantage of?) potential assortative mating ?

– The discovery cohort exists of the "British ancestry" population. It is not clear whether these are genetically homogeneous, or whether they are all of European (or other) ancestry ? Did the author confirm their ancestry genetically? Why were the remaining European ancestry individuals not included in discovery ?

– Why did the authors choose for a two-stage approach ? As they seem to have limited statistical power, a full meta-analysis of all available data, possibly with a more stringent significance threshold, might have been more effective for discovery of new loci, and for replication of previously reported loci.

– The authors performed conditional analyses after already performing many other analyses on identified loci, and then identify several additional independent loci. Clearly, conditional analyses should have been performed immediately following the identification of the 12 loci, such that the additional loci could have been included in the follow-up analyses. Also, more details need to be provided on how analyses were performed in the HLA locus – given the complicate of the HLA locus, this requires analyses beyond the typically GWAS analyses.

– It is disappointing that no mention was made of non-European ancestry populations. While they may have been underpowered on their own, they may have increased statistical power in overall analyses.

– I am surprised by the relatively low statistical power of the current study – studies that include 1 million variants typically identify hundred(s) of genetic loci, even when the narrow-sense heritability is relatively low. The identification of just 12 loci seems disappointing; some discussion on potential reasons of low power and low yield would be informative.

– The authors spend a lot of time/text listing the variants they identified in discovery, which ones of those did or did not replicated, how they compare to previously identified loci and how previously identified loci perform in the current analyses. Readers will get lost, and what is needed is a much more succinct, clear cut section on which loci are robustly associated with lifespan. It is also not clear when variants are indeed considered as replicated and what the significance thresholds are. This results in a cluttered list of replicated and non-replicated variants, and by the end of this section, I had no idea what the main findings were.

– The authors report on a number of analyses that turn out to be side-tracks; i.e. reporting (and discussing) variants that did not reach genome-wide significance or that were not replicated, is misleading to readers. Why perform analyses of which the authors then conclude that findings are biased, to then redo analyses in an unbiased manner ? Simply report the unbiased results/analyses.

– The authors state (also in the title) that the genetic findings (mainly related to the PRS) are "predictive". However, I have a feeling that the word "prediction/predictive" is not used in the correct context. The PRS is "associated" with lifespan, but the authors did not provide data to show that the PRS predicts lifespan (or extreme longevity). The wording (also in the title) (or the analyses) should be corrected.

– The section on implication of causal genes and methylation sites is confusing. Please, provide more details on how the causal genes were identified. In fact, a session on fine-mapping would be informative (e.g. with credible sets).

– Given the vast amount of data, it would also be of interest to have a focus analyses of rare coding variants. Or an analysis of (common and rare) variants and the extremes of the "lifespan" distribution could be informative.

Specific comments:

Title: The title is not clear; no prediction analyses were performed and what is mean with "modern risk"?

Introduction: The introduction is too long and wordy, yet often confusing, as several aspects are not clear. E.g. it should be made clear what the difference is between longevity, lifespan…? The kin-cohort design may need some explanation as well. Heritability estimates are only reported as explained variance from GWAs studies, but what about heritability estimated using family and twin designs.

Results: As mentioned before, it is hard to keep track of which loci are considered "real" and which ones are not. Be more succinct and focus on the key (replicated) findings, no need to provide extensive detail on loci that turn out to be not "real".

Also, there is no need to state over and again that results are not significant because 95%CI overlap with the "0", as that's part of the definition of significance.

The power calculations are not very informative; i.e. the combined analysis had 50% power to detect very common variants (MAF>30%) with an effect size of 0.25 years? 50% is low power. Why not provide the data for 80 or 90% power?

Table 1 is also confusing as it reports many loci that were not replicated. As a main table, readers will want to know which of the loci were truly discovered. Hence, a table should just report the results of the replicated variants; others can be reported in supplementary tables. Also, the legend (of this table and others) is enormous; no need to repeat what has written elsewhere. Clearly, too much redundant information is being reported for the main table as CES P, PDES P and iGWAS p all report the same in this case. Instead of reporting all these statistics in the main table, the authors should consider reporting the most informative results and report others in (very) brief in the text or in supplementary tables.

What is meant with "having converted the recessive effect into an apparent effect for a truly recessive allele"?

The iGWAS needs some explanation in the main text. Based on the highlights in Table 2—source data 2, it seem BMI was not included in the iGWAS? If so, please clarify ? Also, why were significant results for BMI not highlighted?

Stating that the narrow sense heritability was increased "by" 79% is mainly misleading, but also not informative when you do not provide from what value it is increasing. Be straightforward and report from which to which value the narrow-sense heritability is increased.

Provide more details on how cell type and pathway enrichment analyses were performed in the main text.

The so-called "out of sample lifespan predictions" should not be labeled as predictions – no predictive analyses were performed. The analyses reported on in this section seems to be a simple validation of a PRS in another population. Some discussion on why the difference between top and bottom deciles in the Estonian population is smaller would be informative.

Discussion: I would be interested in a greater discussion as to why findings from extreme longevity studies are not (all) replicated in the lifespan studies and vice versa. [No need to speculate on the statistics power, as this can be calculated.] In additional, more insight into why genetic variants do not seems to affect lifespan would be of interest too. The author may consider working with PRS and also estimate genetic correlations.

*Reviewer #2:*

The manuscript by Timmers et al. describes a multi-study GWAS for parental age at death, a trait which has been show to efficiently identify loci for longevity. Although the novelty of this paper is compromised by other similar papers using highly overlapping datasets, the authors should be commended for their rigorous analytical approach and attention to detail. I did however feel at times that clarity of story was sometimes scarified to demonstrate this rigor, and would recommend that the (main) text was re-read and simplified where possible.

I do feel that the heavy focus on lack of replication from one of the previous papers in this area is a bit overstated. Pilling et al. (I'm not connected to this study) did evaluate a polygenic risk score (PRS) in an external dataset, although they were not able to replicate individual variants. Although this current paper does a far better job in that respect, it is still the case that only a very small fraction of the loci presented in main Table 1 replicate in the true sense of the word (i.e P < 0.05 / N). Whilst I definitely don't dispute that replication is important, many large-scale discovery GWAS studies (for example GIANT Locke et al) do not replicate individual SNPs, but may instead evaluate PRS performance in other studies. Ultimately the nature of replication may have more importance when considering what other investigators do with these results downstream. For example, the individual variant level of significance is more important when considering experimental follow-up vs inclusion in a PRS.

Given many of the loci in Table 1 come from the discovery + replication combined analysis, it may be more appropriate to present this as the primary analysis. This would simplify much of the manuscript and allow for a more detailed presentation of the more interesting analyses (some of which were buried in supplemental sections and not described well in the main text). This doesn't preclude first evaluating/replicating the loci previously reported by Pilling et al.

When considering why this study had identified fewer loci than Pilling et al., I noticed that only unrelated individuals were included. This must result in quite a large sample drop, which seems unnecessary. It may also increase novelty to rerun this analysis using the full (v3) UKBB imputation (rather than v2) and include the X-chromosome (I couldn't tell if this had been included).

Can the authors estimate what proportion of the heritable component of lifespan is explained by avoiding disease (i.e identifying protective alleles) vs unexplained mechanisms. This seems key for considering future studies in this area, and I wonder if there are other ways this could be more directly assessed?

*Reviewer #3:*

The paper by Joshi et al. reports the results from a GWAS on parental lifespan, which consist of a discovery (UK Biobank) and replication phase (Lifegen). The authors identified multiple novel loci for parental lifespan when combining the discovery and replication phase. In addition, they were able to replicate several previously identified loci for parental lifespan as well as longevity. They subsequently perform follow-up analyses (including polygenic risk score analyses) to pinpoint the affected pathways as well as relations with other phenotypes and show a relation between parental lifespan and several age-related diseases.

Major comments:

1) Instead of reporting the most significant variant within a locus, it would be more appropriate to report the most-likely causal variant as well as independent genetic variants within each locus. This is common practice in GWAS.

2) The authors performed a meta-analysis on three published longevity GWAS to use for replication of their findings (subsection “Mortality risk factor-informed GWAS (iGWAS)”). They thereby assumed that these three studies looked at the same phenotype, which is actually not the case (one looked at mortality and the other two at longevity (with one using death controls only and the other one dead + alive controls). In addition, the cohorts used in these three GWAS are partly overlapping, so they cannot be analyzed together. Instead the authors should look at the replication separately in each of the studies and update Figure 4 and Figure 4—source data 1 accordingly.

3) If the authors assume that there may be sex-specific effects, why have they not performed a sex-stratified analysis to see if this is indeed the case? They currently only looked at sex-specific effects for their top hits.

4) The out-of-sample lifespan predictions actually highlight an important limitation of using the parental age at death as proxy for an individual's own age at death, since the polygenic risk score works less well in the subjects as compared to their parents. This should be discussed in more detail in the Discussion section.

5) The fact that a higher polygenic risk score results in an increased prevalence of Alzheimer's disease, Parkinson's disease, and prostate and breast cancer shows that the majority of the identified genetic variants are likely not related to healthy ageing, but only to lifespan. Hence, the question arises if this is what we are aiming for with our efforts to unravel the genetics of longevity. This should be discussed in more detail in the Discussion section.

6) I think the authors should mention that the prediction that SESN1 is the gene driving the effect at the FOXO3A locus is likely incorrect, since the functional study show that the variant at this locus has a direct effect on the functioning of FOXO3A itself. Hence, this shows that predictions can be informative but are not always supported by functional evidence.

7) I think the part in the Discussion section about the comparison between lifespan and annuity pricing is farfetched and does not add anything to the manuscript. In addition, I do not agree with the conclusion that the results from the polygenic risk score are "meaningful socially and actuarially", since these results are not that convincing (see comment 6). I would advise to rewrite or even completely remove this section.

Minor Comments:

– The Title of the manuscript is a bad representation of the results described in the manuscript, especially the part "reveal basis in modern risks", which is merely speculative. Hence, the authors should come up with a Title that better carries the load.

– Introduction section: The narrow-sense heritability for longevity mentioned in the study by Kaplanis et al. is 12.2% and since they compared this heritability with the previously published ones, it would be good to also mention that one here (instead of the 16.1%).

– Results paragraph two: Reference 10 should be reference 7.

– I think the word "causal" opening sentence of subsection “Implication of causal genes and methylation sites” is misleading, since the authors are not able to prove causality for their variants with the analyses they performed.

– The numbering of the Supplementary Tables is not in line with the text.

---

## [Author Response]

The major concerns are as follows:– We found the paper was often hard to read and would benefit from substantial re-writing to improve clarity, being more succinct at some points, and providing additional necessary details at other occasions.

With hindsight, we agree. The original version followed too closely our results development process, rather than looking at the end results holistically. We have revised the manuscript substantially, and focused results and discussion as suggested in specific reviewer comments.

– We recommend that the authors limit the discussion around the (non)replication of results, and focus more attention on the subset of loci that show significant association.

Yes, this has been done. Indeed, much of the non-replication aspects have been superseded by the dropping of the multi-stage design (see next editor comment).

– We felt that (as has been shown repeatedly) a one-stage approach – using a more stringent significance threshold – would likely provide more statistical power for discovery, than the current multiple-stage design. It would also simplify and focus the reporting, allowing more in-depth analysis and discussion of each (robustly) associated locus.

We accept this suggestion. As suggested the results do appear robust and the presentation is much more straightforward, i.e. we changed our two-stage discovery and replication into a single stage. To determine an adequate significance threshold, we looked for GWAS studies published in 2018 and found studies only adjusted their threshold when analysing multiple traits or rare variants (see included document). We chose to apply a threshold of 2.5e-8 to account for multiple testing and validate our results using a polygenic risk score. The change affects much of the manuscript, but is most obvious on Table 1.

– We recommend that UKBiobank analyses use appropriate statistical tools to account for population stratification and/or (cryptic) relatedness (e.g. BOLT-LMM);

We agree LMM is often useful in UK Biobank, but judge that the kin-cohort method is an exception. We have now explained this in subections “GWAS” and “Data sources”.

and that any overlap between samples is removed when combining data.

Agreed, this was a misunderstanding on our part. However, as we do not have access to the cohort level data from the meta-analyses performed by other researchers, we have instead adjusted standard errors to reflect the overlap. See subsections “Mortality risk factor-informed GWAS (iGWAS)”, “Candidate SNP replication” and “Replication in extreme long-livedness”.

– We recommend that analyses indicating significant sex-specific effects include both sex-interaction and sex-stratified analyses.

We apologise: our terminology Common Effect Size (CES) and Potentially Different Effect Size (PDES) was confusing. Our approach was to start with sex-stratified analyses (mothers and fathers). The meta-analysis of these results, assuming a single common effect size was thus an unstratified analysis. However, in the PDES analysis, the use of MANOVA, meant that the father and mother traits in the latter analysis was a generalised form of sex-stratification. In particular, an effect on only one sex would be apparent, albeit with an appropriate adjustment to P value to reflect the implicit multiple testing. This approach has the advantage of permitting antagonistic pleiotropy across sexes or (as we in fact observe) different effect sizes in the same direction to be combined, with proper adjustment for the implicit multiple correlated tests, akin to Fishers method for combining P values, for independent tests. We have amended the analysis names to "Sex Common Effect Size" and "Sex Specific Effect Size"n i.e. stratified in the latter case – See subsections “Genome-wide association analysis”, paragraph two of “GWAS” and “UK Biobank Genome-Wide Association Study”.

– We felt strongly that there was an opportunity for more in-depth discussion to increase the informativeness of the study: see reviewer comments for suggestions.

Thank you. We have expanded our discussion significantly.

– An important topic that has not been fully worked out relates to the proportion of the genetic basis of longevity that is explained by avoiding disease vs other mechanisms; i.e. is longevity simply about avoiding disease ?

This is a good question, but one which our data and results can only provide a glimpse of the answer to. We have calculated the proportion of heritability explained by the disease SNPs – see paragraph six of “Disease and lifespan”. This is limited. i.e. there is much missing h2 and it is as yet not possible to say its split between disease and aging.

Separate reviews (please include a response on each point):Reviewer #1:[…] The study design is confusing and seems generally inefficient.– For example, why were >70,000 related individuals (i.e. >17% of the population) removed from analyses – methods exist to use the information of related individuals (e.g. BOLT-LMM).

Please, see above response to the editor.

– The kin-cohort is an interesting design; how does deal with (or take advantage of?) potential assortative mating?

Yes, comment added in discussion, see paragraph seven.

– The discovery cohort exists of the "British ancestry" population. It is not clear whether these are genetically homogeneous, or whether they are all of European (or other) ancestry? Did the author confirm their ancestry genetically? Why were the remaining European ancestry individuals not included in discovery?

Although somewhat superseded by the dropping of the two-stage approach, we have clarified ancestry in subsection “GWAS” and “Data sources”, and do include European ancestry.

– Why did the authors choose for a two-stage approach? As they seem to have limited statistical power, a full meta-analysis of all available data, possibly with a more stringent significance threshold, might have been more effective for discovery of new loci, and for replication of previously reported loci.

Accepted, please see the editor response above.

– The authors performed conditional analyses after already performing many other analyses on identified loci, and then identify several additional independent loci. Clearly, conditional analyses should have been performed immediately following the identification of the 12 loci, such that the additional loci could have been included in the follow-up analyses. Also, more details need to be provided on how analyses were performed in the HLA locus – given the complicate of the HLA locus, this requires analyses beyond the typically GWAS analyses.

In line with the previous comment on multiple, potentially biased analyses, we have replaced the PheWAS with a disease lookup in the GWAS catalog and PhenoScanner. This new analysis looks for associations with multiple SNPs in the same locus, and as such including conditional SNPs is no longer relevant.

– It is disappointing that no mention was made of non-European ancestry populations. While they may have been underpowered on their own, they may have increased statistical power in overall analyses.

Yes we could have done this, but the non-European ancestry components were very small in UK Biobank and initial analyses on these sub-cohorts appeared noisy. Trans-ethnic meta-analysis was therefore beyond the scope of this work.

– I am surprised by the relatively low statistical power of the current study – studies that include 1 million variants typically identify hundred(s) of genetic loci, even when the narrow-sense heritability is relatively low. The identification of just 12 loci seems disappointing; some discussion on potential reasons of low power and low yield would be informative.

Yes, we were surprised too, although a little general reasoning has suggested the result is reasonable. We have expanded discussion paragraph five and six, concluding that due to low heritability and the indirect use of parent genotypes, a further order of magnitude increase in sample size is needed, to attain granularity of results available in other GWAMA.

– The authors spend a lot of time/text listing the variants they identified in discovery, which ones of those did or did not replicated, how they compare to previously identified loci and how previously identified loci perform in the current analyses. Readers will get lost, and what is needed is a much more succinct, clear cut section on which loci are robustly associated with lifespan. It is also not clear when variants are indeed considered as replicated and what the significance thresholds are. This results in a cluttered list of replicated and non-replicated variants, and by the end of this section, I had no idea what the main findings were.

We apologise for this. Adopting the one stage approach and removing the skewed downstream analyses has simplified these issues. We have replaced the UK Biobank PheWAS analysis with an unbiased GWAS catalog lookup, using only SNPs passing our stringent genome-wide significance threshold (see subsection “Disease and lifespan”). We have limited the analyses investigating specific loci to SNPs which are either discovered or replicated in our study (for examples, see subsection “Genome-wide association analysis”).

– The authors report on a number of analyses that turn out to be side-tracks; i.e. reporting (and discussing) variants that did not reach genome-wide significance or that were not replicated, is misleading to readers. Why perform analyses of which the authors then conclude that findings are biased, to then redo analyses in an unbiased manner? Simply report the unbiased results/analyses.

Agreed. We have replaced the UK Biobank PheWAS analysis with independent GWAS catalog lookup, using only SNPs passing our stringent genome-wide significance threshold (see “Disease and lifespan”). We have limited the analyses investigating specific loci to SNPs which are either discovered or replicated in our stud.

– The authors state (also in the title) that the genetic findings (mainly related to the PRS) are "predictive". However, I have a feeling that the word "prediction/predictive" is not used in the correct context. The PRS is "associated" with lifespan, but the authors did not provide data to show that the PRS predicts lifespan (or extreme longevity). The wording (also in the title) (or the analyses) should be corrected.

We see both sides of this issue. We have not created predicted ages at death and compared them with actual ages at death, not least because of censoring. On the other hand our scores are predictive of survival (e.g. the KM curves). Although "associated" with survival would be accurate too, we are confident that this association is also predictive. We have moved to "associate" and "explain" in general.

– The section on implication of causal genes and methylation sites is confusing. Please, provide more details on how the causal genes were identified. In fact, a session on fine-mapping would be informative (e.g. with credible sets).

We have now included the analysis method used to implicate causal genes in the main text (subsection “Causal genes and methylation sites”) and summarised the results more clearly.

– Given the vast amount of data, it would also be of interest to have a focus analyses of rare coding variants. Or an analysis of (common and rare) variants and the extremes of the "lifespan" distribution could be informative.

We agree this would be an interesting study, perhaps along the lines of https://www.biorxiv.org/content/early/2018/09/04/407981. However, it is beyond the scope of present work.

Specific commentsTitle: The title is not clear; no prediction analyses were performed and what is mean with "modern risk"?

Accepted. We have revised the title along these lines. "Genomics of 1 million parent lifespans implicates novel pathways and common diseases and distinguishes survival chances"

Introduction: The introduction is too long and wordy, yet often confusing, as several aspects are not clear. E.g. it should be made clear what the difference is between longevity, lifespan…? The kin-cohort design may need some explanation as well. Heritability estimates are only reported as explained variance from GWAs studies, but what about heritability estimated using family and twin designs.

Accepted. We revised much of the Introduction, extending discussion of heritability estimates, lifespan vs. longevity, and the kin-cohort method, while reducing overall wordiness and improving clarity.

Results: As mentioned before, it is hard to keep track of which loci are considered "real" and which ones are not. Be more succinct and focus on the key (replicated) findings, no need to provide extensive detail on loci that turn out to be not "real".

Accepted. Our single-phase approach has simplified these issues. For examples, see paragraph two and three of “Genome-wide association analysis”.

Also, there is no need to state over and again that results are not significant because 95%CI overlap with the "0", as that's part of the definition of significance.

Accepted. For examples, see subsection “Genome-wide association analysis”.

The power calculations are not very informative; i.e. the combined analysis had 50% power to detect very common variants (MAF>30%) with an effect size of 0.25 years? 50% is low power. Why not provide the data for 80 or 90% power?

Accepted. Our initial attempt was to show for variants like this we had some (not high) power. However, the conventional calculation with 80% power is perhaps more readily understood and so we now take that approach in paragraph two of subsection “Genome-wide association analysis”.

Table 1 is also confusing as it reports many loci that were not replicated. As a main table, readers will want to know which of the loci were truly discovered. Hence, a table should just report the results of the replicated variants; others can be reported in supplementary tables.

Accepted. Table 1 is now much simpler: see table and legend of Figure 10—figure supplement 1.

Also, the legend (of this table and others) is enormous; no need to repeat what has written elsewhere. Clearly, too much redundant information is being reported for the main table as CES P, PDES P and iGWAS p all report the same in this case. Instead of reporting all these statistics in the main table, the authors should consider reporting the most informative results and report others in (very) brief in the text or in supplementary tables.

Accepted. We have reduced the legends of some display items (e.g. Figure 7), but this is balanced by *eLife*’s request to include relevant methodological information in figure and table legends.

What is meant with "having converted the recessive effect into an apparent effect for a truly recessive allele"?

This is now explained more fully in the final paragraph of subsection “Genome-wide association analysis”. Essentially an additive GWAS can detect recessive effects (albeit at reduced power) and we have calculated the corresponding effect size.

The iGWAS needs some explanation in the main text. Based on the highlights in Table 2—source data 2, it seem BMI was not included in the iGWAS? If so, please clarify ? Also, why were significant results for BMI not highlighted?

Accepted. We expanded the explanation in the main text (subsection “Mortality risk factor-informed GWAS (iGWAS)”). BMI was included in the analysis but was mistakenly not highlighted. This has been added.

Stating that the narrow sense heritability was increased "by" 79% is mainly misleading, but also not informative when you do not provide from what value it is increasing. Be straightforward and report from which to which value the narrow-sense heritability is increased.

Accepted and actioned. See subsection “Causal genes and methylation sites”.

Provide more details on how cell type and pathway enrichment analyses were performed in the main text.

Accepted. See subsection “Cell type and pathway enrichment”.

The so-called "out of sample lifespan predictions" should not be labeled as predictions – no predictive analyses were performed. The analyses reported on in this section seems to be a simple validation of a PRS in another population. Some discussion on why the difference between top and bottom deciles in the Estonian population is smaller would be informative.

Accepted. We have now moved to "associate" and discussed the reduced explanatory power in Estonia. See paragraph fifteen of the Discussion section.

Discussion: I would be interested in a greater discussion as to why findings from extreme longevity studies are not (all) replicated in the lifespan studies and vice versa. [No need to speculate on the statistics power, as this can be calculated.] In additional, more insight into why genetic variants do not seems to affect lifespan would be of interest too. The author may consider working with PRS and also estimate genetic correlations.

Yes, done in Discussion paragraph eleven. Broadly we consider the traits as highly overlapping (at least genetically). We find that lifespan SNPs are longevity SNPs (Figure 4) and that some reported longevity SNPs that are not lifespan SNPs may be false positives, or less plausibly only have effects delayed well beyond those of APOE e4.

Reviewer #2:The manuscript by Timmers et al. describes a multi-study GWAS for parental age at death, a trait which has been show to efficiently identify loci for longevity. Although the novelty of this paper is compromised by other similar papers using highly overlapping datasets, the authors should be commended for their rigorous analytical approach and attention to detail. I did however feel at times that clarity of story was sometimes scarified to demonstrate this rigor, and would recommend that the (main) text was re-read and simplified where possible.I do feel that the heavy focus on lack of replication from one of the previous papers in this area is a bit overstated. Pilling et al. (I'm not connected to this study) did evaluate a polygenic risk score (PRS) in an external dataset, although they were not able to replicate individual variants. Although this current paper does a far better job in that respect, it is still the case that only a very small fraction of the loci presented in main Table 1 replicate in the true sense of the word (i.e P < 0.05 / N). Whilst I definitely don't dispute that replication is important, many large-scale discovery GWAS studies (for example GIANT Locke et al) do not replicate individual SNPs, but may instead evaluate PRS performance in other studies. Ultimately the nature of replication may have more importance when considering what other investigators do with these results downstream. For example, the individual variant level of significance is more important when considering experimental follow-up vs inclusion in a PRS.

Accepted. We have moved to a single stage approach, with a lower P value threshold, and an approach akin to testing a PRS in an independent population, but using summary statistics from the second population, an approach familiar from two sample mendelian randomisation. We hope this is a good compromise, as well as a simpler presentation: whilst well powered for the score as a whole, and underpowered variant by variant, we believe the approach (e.g. Figure 4) does give a sense of the degree of confidence of individual variants. This reasoning has been added in subsection “Mortality risk factor-informed GWAS (iGWAS)”.

Given many of the loci in Table 1 come from the discovery + replication combined analysis, it may be more appropriate to present this as the primary analysis. This would simplify much of the manuscript and allow for a more detailed presentation of the more interesting analyses (some of which were buried in supplemental sections and not described well in the main text). This doesn't preclude first evaluating/replicating the loci previously reported by Pilling et al.

Accepted. See response to editor above, and Figure 2, which evaluates prior findings in our independent datasets.

When considering why this study had identified fewer loci than Pilling et al., I noticed that only unrelated individuals were included. This must result in quite a large sample drop, which seems unnecessary. It may also increase novelty to rerun this analysis using the full (v3) UKBB imputation (rather than v2) and include the X-chromosome (I couldn't tell if this had been included).

We believe our study is more conservative than Pilling et al. Using our one stage approach, we now have a bigger sample size, despite restricting to unrelated samples, rather than LMM (see editor response). We have set a significance threshold that is conservative with respect to multiple testing, in accordance with editorial suggestion. Nonetheless we do accept some of Pilling et al's hits are likely real, and try to present further evidence in figures. Extension of analysis to the new UK Biobank a imputation is beyond the scope of this manuscript.

Can the authors estimate what proportion of the heritable component of lifespan is explained by avoiding disease (i.e identifying protective alleles) vs unexplained mechanisms. This seems key for considering future studies in this area, and I wonder if there are other ways this could be more directly assessed?

Agreed. See response to editor above and reviewer 1 final comment.

Reviewer #3:The paper by Joshi et al. reports the results from a GWAS on parental lifespan, which consist of a discovery (UK Biobank) and replication phase (Lifegen). The authors identified multiple novel loci for parental lifespan when combining the discovery and replication phase. In addition, they were able to replicate several previously identified loci for parental lifespan as well as longevity. They subsequently perform follow-up analyses (including polygenic risk score analyses) to pinpoint the affected pathways as well as relations with other phenotypes and show a relation between parental lifespan and several age-related diseases.Major comments:1) Instead of reporting the most significant variant within a locus, it would be more appropriate to report the most-likely causal variant as well as independent genetic variants within each locus. This is common practice in GWAS.

Our SOJO analysis has reported independent genetic variants within each locus, and following your comment, we attempted a Bayesian fine-mapping analysis using JAM to create credible sets, but this was underpowered. As such, we have chosen to report the index SNP as most-likely causal. We have changed the wording in table legends to reflect lead variants are index SNPs.

2) The authors performed a meta-analysis on three published longevity GWAS to use for replication of their findings (subsection “Mortality risk factor-informed GWAS (iGWAS)”). They thereby assumed that these three studies looked at the same phenotype, which is actually not the case (one looked at mortality and the other two at longevity (with one using death controls only and the other one dead + alive controls). In addition, the cohorts used in these three GWAS are partly overlapping, so they cannot be analyzed together. Instead the authors should look at the replication separately in each of the studies and update Figure 4 and Figure 4—source data 1 accordingly.

We are sorry not to have spotted the sample overlap issue and thank you for bringing it to our attention. We have now adjusted for it using an adjustment to the standard error, in the meta-analysis. We recognise that the traits are somewhat heterogeneous, but we are trying to replicate a distinct trait anyway. By recalibrating each study, we have made a best estimate meta-analysis, and most importantly, the meta-analysed effect will have the required behaviours under the null hypothesis, due to the recalibrated SE. See Figure 4—source data 1.

3) If the authors assume that there may be sex-specific effects, why have they not performed a sex-stratified analysis to see if this is indeed the case? They currently only looked at sex-specific effects for their top hits.

Agreed sex stratification is desirable, but we did do this (and apologise that it was unclear). See response to the editor above.

4) The out-of-sample lifespan predictions actually highlight an important limitation of using the parental age at death as proxy for an individual's own age at death, since the polygenic risk score works less well in the subjects as compared to their parents. This should be discussed in more detail in the Discussion section.

We agree. We have drawn this point out more in paragraph fifteen of the Discussion section. And note the reduction is only 20% – i.e. the signal is far from lost and across the deciles the hazard ratio is the same in both generations.

5) The fact that a higher polygenic risk score results in an increased prevalence of Alzheimer's disease, Parkinson's disease, and prostate and breast cancer shows that the majority of the identified genetic variants are likely not related to healthy ageing, but only to lifespan. Hence, the question arises if this is what we are aiming for with our efforts to unravel the genetics of longevity. This should be discussed in more detail in the Discussion section.

Yes, see paragraph eleven of the Discussion section.

6) I think the authors should mention that the prediction that SESN1 is the gene driving the effect at the FOXO3A locus is likely incorrect, since the functional study show that the variant at this locus has a direct effect on the functioning of FOXO3A itself. Hence, this shows that predictions can be informative but are not always supported by functional evidence.

While we agree the functional study has shown genetic variation at this locus influences FOXO3 expression, SMR-HEIDI does not find a link between FOXO3 expression and lifespan in our own data. Differential FOXO3 expression does not preclude a role for SESN1, which is connected to the same promotor and may thus also be differentially expressed. We have added this line of reasoning to the discussion and stressed the need for follow-up work.

7) I think the part in the Discussion section about the comparison between lifespan and annuity pricing is farfetched and does not add anything to the manuscript. In addition, I do not agree with the conclusion that the results from the polygenic risk score are "meaningful socially and actuarially", since these results are not that convincing (see comment 6). I would advise to rewrite or even completely remove this section.

We accept the phrase "meaningful" was subjective. However, we do feel we evidence that a 14% difference in price between the two pools is material – as two insurers playing solely in either of those pools would see profits double or almost wiped out (8.9% +/- 7%). Even if the 20% dilution observed across the whole spectrum applies, we would see an 11% distinction in profitability across the pools. In any case, there is increasing interest in the association of polygenic scores with outcomes and their use in potential screening programmes for example Khera et al. PMID: 30104762. Although it is too early to evidence in publications, that work in particular has to our knowledge attracted interest from the actuarial profession including active follow-up work. Furthermore, the present work will be presented by Dr Joshi FoIFA to a sessional meeting of the UK actuarial profession on 21 January 2019. To contextualise, the score performs in line with the UK insurers leading existing method of annuity underwriting – postcodes. We also have clarified the deciles hazard (only) remains constant and tried to be clear we are not advocating genetic testing for insurance, but highlighting the need for regulation to be updated.

Minor Comments:– The Title of the manuscript is a bad representation of the results described in the manuscript, especially the part "reveal basis in modern risks", which is merely speculative. Hence, the authors should come up with a Title that better carries the load.

Accepted. We have revised the title on the "predicts" and "risks" aspects.

– Introduction section: The narrow-sense heritability for longevity mentioned in the study by Kaplanis et al. is 12.2% and since they compared this heritability with the previously published ones, it would be good to also mention that one here (instead of the 16.1%).

Agreed. This has been updated.

– Results paragraph two: Reference 10 should be reference 7.

Thank you for spotting this. It has now been superseded by rewriting of the Introduction.

– I think the word "causal" opening sentence of subsection “Implication of causal genes and methylation sites” is misleading, since the authors are not able to prove causality for their variants with the analyses they performed.

We have weakened the wording slightly, and agree with the reviewer that we have not shown the variants are causal, however we are confident the loci are causal, and suggest the genes are causal through use of mendelian randomisation on the genes/loci. Nonetheless, we have weakened the wording slightly, as MR does rely on the no horizontal pleiotropy assumption, which is highly plausible here, but not provable even for in cis action, and is still valid for variants in LD with the causal variant. Subsection “Causal genes and methylation sites”.

– The numbering of the Supplementary Tables is not in line with the text.

Apologies, we have now corrected this.